# NN-ResDMD: Learning Koopman Representations for Complex Dynamics with Spectral Residuals

## Abstract

Analyzing long-term behaviors in high-dimensional nonlinear dynamical systems remains a significant challenge. The Koopman operator framework has emerged as a powerful tool to address this issue by providing a globally linear perspective on nonlinear dynamics. However, existing methods for approximating the Koopman operator and its spectral components, particularly in large-scale systems, often lack robust theoretical guarantees. Residual Dynamic Mode Decomposition (ResDMD) introduces a spectral residual measure to assess the convergence of the estimated Koopman spectrum, which helps filter out spurious spectral components. Nevertheless, it depends on pre-computed spectra, thereby inheriting their inaccuracies. To overcome its limitations, we introduce the Neural Network-ResDMD (NN-ResDMD), a method that directly estimates Koopman spectral components by minimizing the spectral residual. By leveraging neural networks, NN-ResDMD automatically identifies the optimal basis functions of the Koopman invariant subspace, eliminating the need for manual selection and improving the reliability of the analysis. Experiments on physical and biological systems demonstrate that NN-ResDMD significantly improves both accuracy and scalability, making it an effective tool for analyzing complex dynamical systems.

## 1 Introduction

In the study of complex dynamical systems, a critical challenge lies in accurately extracting and analyzing long-term behavior in high-dimensional nonlinear systems. Various data-driven methods (Brunton & Kutz, 2019; Schetzen, 2006; Wiggins, 2003; Slotine & Li, 1991; Lan & Mezić, 2013; Mezić, 2005) have been developed to address this challenge, with the Koopman operator (Koopman, 1931; Koopman & Neumann, 1932) framework emerging as a powerful tool due to its ability to globally linearize nonlinear systems. Unlike local linearization methods (Hartman, 1960; Grobman, 1959), which approximate dynamics near fixed points, the Koopman operator transforms the entire system into a linear form within an infinite-dimensional space, which allows the use of spectral analysis techniques to study complex dynamics.

Despite its promise, practical computational challenges arise from the infinite-dimensional nature of the Koopman operator. Numerical methods such as Extended Dynamic Mode Decomposition (EDMD) (Williams et al., 2015) have been developed to approximate the Koopman operator using a finite set of observables, making it possible to extract dynamic modes from data. However, EDMD lacks theoretical guarantees of convergence and may fail to capture the full Koopman spectrum accurately, particularly in large-scale, complex systems.

To address these limitations, Residual Dynamic Mode Decomposition (ResDMD) (Colbrook & Townsend, 2024) was introduced, offering convergence guarantees through a spectral residual measure that quantifies how well the estimated Koopman spectrum converges to the true spectrum. By assessing convergence, ResDMD eliminates spurious spectral components that do not represent the system's true dynamics, enhancing the reliability of spectral estimation. However, ResDMD primarily filters precomputed spectra rather than directly approximating Koopman spectra, limiting its ability to independently refine spectral estimates.

In this paper, we propose Neural Network-ResDMD (NN-ResDMD), which overcomes this limitation by providing a method to directly compute Koopman eigenpairs by minimizing the spectral residual. Additionally, NN-ResDMD employs neural networks to automatically select basis functions, eliminating the need for manual intervention, a common challenge in EDMD-based methods. Through experiments on both toy models and real-world high-dimensional systems, we demonstrate that NN-ResDMD significantly improves accuracy and scalability, making it a practical and effective tool for analyzing complex dynamical systems.

## 2 PRELIMINARY ON KOOPMAN OPERATOR

Consider a discrete-time dynamical system $(\Omega, \mu)$ governed by a map $F : \Omega \to \Omega$, where $\Omega \subseteq \mathbb{R}^d$ is the state space, and $\mu$ is a probability measure. The evolution of the system is described by:

$$x_{k+1} = F(x_k), \quad k \in \mathbb{Z}^+.$$

The Koopman operator $\mathcal{K}$ acts on observables $g \in L^2(\Omega, \mu)$ as:

$$\mathcal{K}g = g \circ F.$$

Although $F$ is nonlinear, the Koopman operator $\mathcal{K}$ is linear, enabling spectral analysis of the system in the infinite-dimension function space.

A key aspect of modern Koopman operator theory is Koopman Mode Decomposition (KMD) (Mezić, 2005), which represents system dynamics through its spectral components, i.e. the eigenvalues, Koopman modes, and eigenfunctions. The discrete spectrum is particularly important for insights into long-term behavior, such as periodicity and stability. Our analysis emphasizes these spectral components derived from KMD. Specifically, we seek eigenpairs $(\lambda_i, \phi_i)$, where $\lambda_i$ are eigenvalues and $\phi_i$ are the corresponding Koopman eigenfunctions.

One of the most prominent numerical methods to approximate the Koopman operator and its spectral components is the Extended Dynamic Mode Decomposition (EDMD) method, introduced by Williams et al. (2015). In EDMD, a set of observables (dictionary or basis functions) $\mathbf{\Psi} = [\psi_1, \ldots, \psi_{N_K}]$ is selected, and the span of these observables defines the subspace $V_{N_K} := \operatorname{span}\{\psi_i\}_{i=1}^{N_K}$. Snapshots of the system's state are then collected, and the method constructs a finite-dimensional approximation of the Koopman operator by solving a least-squares problem that relates the snapshots of observables. This enables the computation of eigenvalues, eigenfunctions, and Koopman modes. Note that while common choices of dictionary functions are polynomials, Fourier basis, RBF functions, etc., the optimal choice of basis functions is usually unknown a priori and depends heavily on the specific dynamical system.

Given independent and identically distributed data snapshots $\{(x_i, y_i)\}_{i=1}^m$ with $y_i = F(x_i)$, two matrices $\Psi_X$ and $\Psi_Y$ are formed by evaluating the dictionary on the data snapshots:

$$\Psi_X = \begin{bmatrix} \psi_1(x_1) & \ldots & \psi_{N_K}(x_1) \\ \vdots & \ddots & \vdots \\ \psi_1(x_m) & \ldots & \psi_{N_K}(x_m) \end{bmatrix}, \quad \Psi_Y = \begin{bmatrix} \psi_1(y_1) & \ldots & \psi_{N_K}(y_1) \\ \vdots & \ddots & \vdots \\ \psi_1(y_m) & \ldots & \psi_{N_K}(y_m) \end{bmatrix}.$$

EDMD computes the Koopman matrix approximation as $K = \Psi_X^\dagger \Psi_Y$, where $\Psi_X^\dagger$ is the pseudo-inverse of $\Psi_X$. The eigenvalues of $K$ provide approximations of the Koopman operator's spectrum, and the Koopman eigenfunctions $\phi_i$ are approximated as $\phi_i = \mathbf{\Psi} \mathbf{v}_i$, where $\mathbf{v}_i \in \mathbb{C}^{N_K}$ is the $i$-th eigenvector of $K$.

## 3 KOOPMAN OPERATOR LEARNING

While EDMD approximates the Koopman operator, it suffers from spectral pollution as increasing dictionary sizes introduce spurious eigenvalues, obscuring the system's true dynamics. Residual Dynamic Mode Decomposition (ResDMD) (Colbrook & Townsend, 2024) addresses this by filtering out spurious eigenvalues using spectral residuals but relies on precomputed eigenpairs, inheriting inaccuracies from methods like EDMD.

To overcome these limitations, we propose Neural Network-ResDMD (NN-ResDMD), which directly approximates the Koopman operator and its spectrum by minimizing a ResDMD-specific loss function. Using a Feedforward Neural Network (FNN), NN-ResDMD also optimizes dictionary functions for the Koopman invariant subspace, eliminating the need for manual basis selection.

## 3.1 RESDMD REVIEW

Now, suppose we have obtained an eigenpair $(\lambda, \phi)$ of $\mathcal{K}$ from EDMD or other methods (Colbrook, 2023; Baddoo et al., 2021; Alford-Lago et al., 2022a; Schmid, 2010; Tu et al., 2014; Li et al., 2017) where $\lambda \in \mathbb{C}$ and the eigenfunction $\phi$ is expanded in terms of dictionary functions, i.e., $\phi = \boldsymbol{\Psi}\mathbf{v} = \sum_{i=1}^{N_K} \psi_i v_i \in V_{N_K}$ for some $\mathbf{v} \in \mathbb{C}^{N_K}$, where $v_i$ represents weights of the span. Without loss of generality, we consider $\phi$ has been normalized, i.e., $\|\phi\|_2 = 1$. The accuracy of this eigenpair approximation in the ResDMD framework can be measured by computing its *squared relative residual* using the dictionary in the following way:

$$res(\lambda, \phi)^2 := \frac{\int_\Omega |\mathcal{K}\phi(x) - \lambda\phi(x)|^2 d\mu(x)}{\int_\Omega |\phi(x)|^2 d\mu(x)}$$

$$= \sum_{i,j=1}^{N_K} \bar{v}_i \left[ \langle \mathcal{K}\psi_i, \mathcal{K}\psi_j \rangle_\mu - \lambda \langle \psi_i, \mathcal{K}\psi_j \rangle_\mu - \bar{\lambda} \langle \mathcal{K}\psi_i, \psi_j \rangle_\mu + |\lambda|^2 \langle \psi_i, \psi_j \rangle_\mu \right] v_j, \quad (3.1)$$

where $\bar{v}_i, \bar{\lambda}$ denote the complex conjugate of $v_i, \lambda$.

This *squared relative residual* in (3.1) is the theoretical value that measures the distance between $\phi$ and the eigenspace associated with $\lambda$, especially under the assumption that $\lambda$ is in the discrete spectrum of $\mathcal{K}$. To approximate this residual in practice, we apply the Galerkin approximation (Boyd, 2013), which states that as the number of data points $m$ increases, the following limits hold:

$$\lim_{m\to\infty} \frac{1}{m} [\Psi_X^* \Psi_X]_{ij} = \langle \psi_i, \psi_j \rangle_\mu,$$

$$\lim_{m\to\infty} \frac{1}{m} [\Psi_X^* \Psi_Y]_{ij} = \langle \psi_i, \mathcal{K}\psi_j \rangle_\mu, \quad (3.2)$$

$$\lim_{m\to\infty} \frac{1}{m} [\Psi_Y^* \Psi_Y]_{ij} = \langle \mathcal{K}\psi_i, \mathcal{K}\psi_j \rangle_\mu = \langle \psi_i, \mathcal{K}^*\mathcal{K}\psi_j \rangle_\mu,$$

where $*$ denotes complex conjugate. Using this approximation, the *squared relative residual* from (3.1) is approximated as follows (see A.1 for more details):

$$\widehat{res}(\lambda, \phi)^2 := \frac{1}{m}\mathbf{v}^* \left[ \Psi_Y^* \Psi_Y - \lambda(\Psi_X^* \Psi_Y)^* - \bar{\lambda}\Psi_X^* \Psi_Y + |\lambda|^2 \Psi_X^* \Psi_X \right] \mathbf{v}. \quad (3.3)$$

where (3.3), denoted as $\widehat{res}(\lambda, \phi)^2$, represents the approximation of the theoretical value in (3.1).

By definition in (3.1), the residual quantifies the deviation from the spectral property, measuring how far the estimated eigenpair is from the true spectrum. In practice, (3.3) is calculated for all precomputed eigenpairs, retaining those with residuals below a threshold. However, while residuals help filter and select valid eigenpairs, they do not improve the accuracy of eigenpair estimation.

## 3.2 NEURAL NETWORK-RESDMD

**General framework** In this section, we present the Neural Network-ResDMD (NN-ResDMD) framework, designed to compute the eigenpairs of the Koopman operator directly using ResDMD-based spectral residuals, as illustrated in Figure 1. The method first determines the optimal dictionary functions by minimizing the *total residual* $J := \sum_{i=1}^{N_K} \widehat{res}(\lambda_i, \phi_i)^2$, over all computed eigenpairs $\{(\lambda_i, \phi_i)\}_{i=1}^{N_K}$. The spectral residual directly impacts the finite-dimensional projection of the Koopman operator and our method minimizes this residual to ensure the learned basis functions adequately capture the Koopman dynamics. This approach allows the construction of the Koopman operator matrix $\tilde{K}$ without relying on external methods or post-processing. Equation (3.5) enables NN-ResDMD to compute eigenpairs directly, improving accuracy compared to ResDMD, which relies on filtering precomputed results from other methods.

In this framework, neural networks parameterize the dictionary functions $\boldsymbol{\Psi}(x;\theta)$, where $\theta$ represents the network parameters. By minimizing the spectral residual $J$, this approach directly optimizes the dictionary functions towards better approximation of Koopman spectral components, which ensures the learned operator captures the underlying spectral properties of the dynamical system. This is fundamentally different from traditional methods like EDMD, which focus on minimizing prediction errors in the observable space without explicitly considering spectral accuracy. The neural network architecture serves as a flexible function approximator, that allows the framework to adaptively learn the optimal dictionary that minimizes spectral residuals, thereby producing more accurate and reliable Koopman spectral decompositions. This spectral-oriented optimization improves the accuracy of eigenvalues approximations and enhances the quality of the computed eigenfunctions, which leads to better characterization of the system's dynamic behavior.

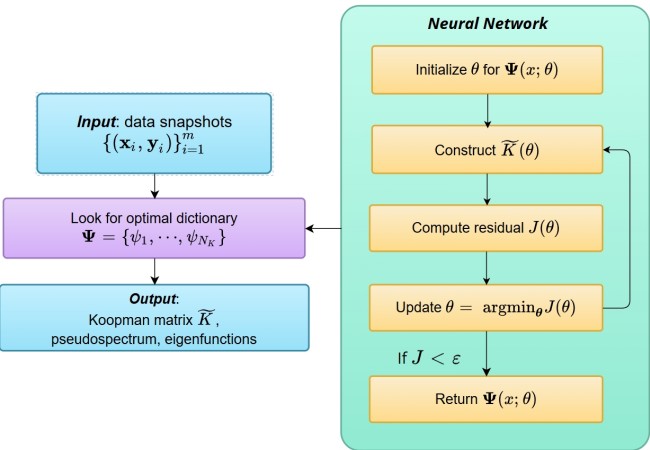

Figure 1: (Left) The classical ResDMD and (Right) the Neural Networks based ResDMD methods

**From Residual to NN** This section explains how neural networks are integrated into the ResDMD framework. In ResDMD, the *squared relative residual* approximation (3.3) measures how well a computed eigenpair fits the dataset. If the Koopman matrix $K$ is well-approximated by the projected Koopman operator $\mathcal{K}_{N_K}$, the *total residual $J$* should approach zero as more data is provided. Thus, $J$ can be used as a loss function, and the optimal Koopman matrix $\tilde{K}$ is obtained by minimizing:

$$J = \sum_{i=1}^{N_K} \widehat{res}(\lambda_i, \phi_i)^2. \tag{3.4}$$

which is equivalent to minimization the following (See A.2 for more details):

$$J = \frac{1}{m}\|(\Psi_Y - \Psi_X K)V\|_F^2 \tag{3.5}$$

where $V$ is a matrix in which each column is an eigenvector $\mathbf{v}_i$ of Koopman matrix $K$. Thus, with a fixed dictionary function $\boldsymbol{\Psi}$, the explicit form for the optimal Koopman matrix $\tilde{K}$ can be directly computed as

$$\tilde{K} = G^\dagger A \tag{3.6}$$

where $G = \frac{1}{m}\Psi_X^*\Psi_X, A = \frac{1}{m}\Psi_X^*\Psi_Y$.

**Remark.** *Typically a regularization term is needed to enhance stability. Here we add a small perturbation, i.e., $\tilde{K} = (G + \sigma I)^{-1}A$ for some small number $\sigma > 0$.*

As shown in (3.6), NN-ResDMD provides an explicit expression for $\tilde{K}$ given the *optimal* dictionary function $\boldsymbol{\Psi}$, allowing for the direct computation of Koopman eigenpairs. The optimization problem in Equation 3.5 is to minimize the error along the eigen-basis, in contrast to the optimization problem $\|\Psi_Y - \Psi_X K\|_F^2$ for EDMD, thereby yielding different *optimal* $\boldsymbol{\Psi}$ compared to EDMD. Therefore, although the $K$ update procedure appear identical to the EDMD approach, they originate from different theoretical foundations and serve different optimization purposes. Additionally, it

automatically optimizes basis functions using neural networks, removing the need for manual selection. Since NN-ResDMD is based on the ResDMD framework, it also retains the theoretical convergence guarantees that EDMD lacks: EDMD has convergence results under strong assumptions, such as requiring the Koopman operator to be bounded (Assumption 2 in Korda & Mezić (2018)), ResDMD requires only that the operator is closed and densely defined.

In NN-ResDMD, neural networks parameterize the dictionary functions $\Psi(x; \theta)$ to minimize the *total residual* $J(\theta)$, as defined in (3.4). The feedforward neural network generates the dictionary functions based on data snapshots, and the total residual is given by:

$$J(\theta) = \frac{1}{m} \|(\Psi_Y(\theta) - \Psi_X(\theta)K(\theta))V(\theta)\|_F^2 \tag{3.7}$$

where $K(\theta)$ and $V(\theta)$ depend on $\theta$. The Koopman matrix $\tilde{K}(\theta)$ is computed as:

$$\tilde{K}(\theta) = G(\theta)^\dagger A(\theta) \tag{3.8}$$

with $G(\theta) = \frac{1}{m}\Psi_X(\theta)^*\Psi_X(\theta)$ and $A(\theta) = \frac{1}{m}\Psi_X(\theta)^*\Psi_Y(\theta)$.

The algorithm alternates between updating $K(\theta)$ via least squares and optimizing $\theta$ using gradient descent until $J(\theta)$ converges, yielding the approximated Koopman spectrum and optimized dictionary functions. While it is possible to optimize both $K(\theta)$ and $J(\theta)$ simultaneously, as done in Takeishi et al. (2017) and Otto & Rowley (2019), our separate procedure ensures computational efficiency and numerical stability compared to the coupled optimization case.

**Computing Algorithm** In our neural networks implementation, we include some non-trainable basis outputs to enhance the dictionary functions. Specifically, we add a vector of ones and the coordinates of the state space as non-trainable basis in the output layer, which help avoid trivial solutions, i.e., $J = 0$ for some initial $\theta$. For the network architecture, we build a three-layer Feedforward Network where each hidden layer size can be specified during training. We use the hyperbolic tangent (tanh) function as the activation function for the hidden layers. In terms of optimization, we employ the Adam optimizer for updating the network parameters. Adam is particularly well-suited for this task due to its ability to adapt the learning rate for each parameter, which can lead to faster convergence in the alternating optimization process between the network parameters and the Koopman matrix. The computing steps are illustrated in the following Algorithm 1.

---

**Algorithm 1:** NN-ResDMD

**Input:** Dataset $X, Y$, number of observables $N_K$, learning step $\delta$, regularization parameter $\sigma$, loss function threshold $\epsilon > 0$, grid points $\{z_1, \ldots z_{n_z}\}$.

1. Initialize $\theta$, thus initializing $\Psi(\theta)$ ;
2. Compute $\tilde{K}(\theta)$ and its eigenvector matrix $V(\theta)$ ;
3. **while** $J(\theta) > \epsilon$ **do**
4.      Update $\theta = \theta - \delta\nabla_\theta J(\theta)$ ;
5.      Compute $G(\theta) = \frac{1}{m}\Psi_X^*\Psi_X, A(\theta) = \frac{1}{m}\Psi_X^*\Psi_Y$;
6.      Update $\tilde{K}(\theta) = (G(\theta) + \sigma I)^{-1}A(\theta)$ and $V(\theta)$ ;

**Output:** $\tilde{K}(\theta)$, eigenpairs $\{(\lambda_i, \phi_i = \Psi\mathbf{v}_i)\}_{i=1}^{N_K}$ and pseudospectrum $\{z_j : \tau_j < \varepsilon\}$.

---

While the practical advantages of NN-ResDMD are demonstrated through experiments, it is important to note its computational demands. The algorithm's computational complexity stems primarily from its iterative optimization process. Each iteration involves a gradient descent update with complexity scaling linearly with both system dimensionality and neural network parameters. Although individual gradient steps are computationally lightweight for standard network architectures, the algorithm's efficiency issue lies in its repeated least-squares optimizations. Compared to standard single least-squares computation as in most numerical algorithms, NN-ResDMD requires multiple iterations to achieve convergence, with stochastic gradient descent methods showing a theoretical $O(1/n)$ convergence rate. However, the method's nonlinear optimization nature also presents challenges for establishing concrete convergence bounds and error estimates.

If the continuous spectrum of the Koopman operator is of interest, following the ResDMD paper's idea, we can scan candidate spectrum values within a grid in the complex plane using the residuals.

Specifically, we compute $\tau_j = \min_{\mathbf{v}_i \in \mathbb{C}^{N_k}} \widehat{res}(z_j, \mathbf{\Psi}(\theta)\mathbf{v}_i)$, where $\tau_j$ is the minimum residual for a grid point $z_j \in \mathbb{C}$. The approximated whole spectrum containing the continuous spectrum is then given by $\{z_j : \tau_j < \varepsilon\}$. More details can be found in (Colbrook & Townsend, 2024).

While the practical advantages of NN-ResDMD are demonstrated through experiments, it's also worth noting that the method has theoretical underpinnings (Haykin, 2009; Weinan et al., 2019) that support its convergence properties. A brief discussion on the convergence aspects of NN-ResDMD, leveraging existing results from approximation theory in Barron spaces, is provided in Appendix A.3. This discussion offers insights into how the neural network component of NN-ResDMD contributes to its effectiveness in approximating complex dynamical systems.

## 4 APPLICATION IN PHYSICAL AND BIOLOGICAL SYSTEMS

In this chapter, we present three examples demonstrating the effectiveness of NN-ResDMD in estimating the key quantities of Koopman Mode Decomposition (KMD): spectrum, eigenfunctions, and Koopman modes. In the first low-dimensional example of a classical pendulum system, our method requires significantly fewer dictionary observables than (Colbrook & Townsend, 2024, Section 4.3.1, Section 6.3) to compute the Koopman spectrum. The second high-dimensional example on turbulence highlights our method's ability to detect acoustic vibrations and distinguish the pressure field through Koopman modes. The third example, a real-world high-dimensional neural system, compares NN-ResDMD with three popular methods—Hankel-DMD (Arbabi & Mezic, 2017), EDMD with RBF basis, and kernelized-ResDMD (Colbrook & Townsend, 2024)—and demonstrates its superiority in identifying and clustering latent dynamic structures. Together, these examples showcase NN-ResDMD's performance across diverse systems and comprehensively evaluate its capabilities.

Specifically, in all three experiments, we compare NN-ResDMD with Hankel-DMD, a theoretically grounded method that analyzes dynamical systems using time-delayed state measurements (see Appendix A.7.1 for a justification). Although its performance rivals NN-ResDMD in the simple pendulum system, it fails to capture key dynamics in higher-dimensional systems.

### 4.1 PENDULUM

The pendulum system is a measure-preserving system due to its Hamiltonian nature, which theoretically implies its whole spectrum lies on the unit circle. For its dynamical behaviors, if the initial position of the pendulum is sufficiently far from the peak and the initial angular speed sufficiently small, the pendulum will oscillate; otherwise, the pendulum will pass the peak and rotate. In other words, this complex system exhibits two types of dynamical behaviors: rotation and oscillation. Here we simulate two cases with different numbers of initial points. We choose 90 and 240 initial points uniformly in the domain $[-\pi, \pi]_{per} \times [-15, 15]$. Each point evolves 1000 steps with a step size of 0.5. Thus, the total data size in each set is approximately $9 \times 10^4$ and $2.4 \times 10^5$, respectively.

As shown in Figure 2, only $N_K = 300$ observables are needed to approximate the full spectrum, significantly fewer than the nearly 1000 required in (Colbrook & Townsend, 2024, Section 4.3.1) for the same data size. Even with a much larger data size (Figure 3), the required observables remain small ($N_K = 350$), demonstrating the robustness of efficient observables across data sizes.

As shown in Figure 4, we compare our method to four approaches: EDMD, EDMD with Dictionary Learning (EDMD-DL), Hankel-DMD, and ResDMD, on the dataset with 90 initial points to compute the Koopman matrix and its corresponding spectral information. The first three methods (EDMD (Williams et al., 2015), EDMD-DL (Li et al., 2017), and Hankel-DMD are limited to computing eigenvalues associated with the point spectrum. In these experiments, both EDMD and ResDMD use the hyperbolic cross approximation with Hermite functions up to order 15 and Fourier functions up to order 20. Hankel-DMD uses a time delay of 150. While Hankel-DMD yields accurate eigenvalues and shows points near the unit circle which matches the ground truth for this Hamiltonian system, it only captures the point spectrum(eigenvalues) and misses the entire spectrum(eigenvalues + continuous spectrum). It also still suffers from spectral pollution and requires careful tuning of the time delay parameter. With 300 basis functions, ResDMD is still unable to fully capture the entire spectrum, i.e., the unit circle, due to the insufficient number of basis functions. In the original ResDMD work, 964 basis functions using a hyperbolic cross approximation of order 100 were required to adequately cover the spectrum with a dataset of the same size (Colbrook & Townsend,

2024, Section 4.3.1). Our method, in contrast, represents the results using shaded areas that show the pseudospectrum, which is a key feature that can capture the whole spectrum. While the shaded area may appear broad, this actually demonstrates our method's ability to detect the complete spectrum. This radius of the shaded region accounts for computational uncertainties, as exact spectrum computation is computationally impossible. Theoretically, ResDMD guarantees that as this error tolerance(radius of the shaded region) approaches zero, the pseudospectrum converges to the true spectrum (the unit circle in this case) without spectral pollution. This comparison demonstrates that NN-ResDMD, even with only 300 basis functions, outperforms all four classical methods in terms of capturing the complete spectrum with greater accuracy and fewer basis functions.

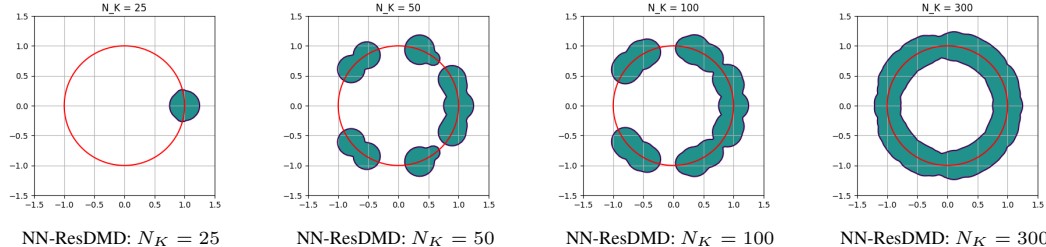

NN-ResDMD: $N_K = 25$    NN-ResDMD: $N_K = 50$    NN-ResDMD: $N_K = 100$    NN-ResDMD: $N_K = 300$

Figure 2: The four plots depict the spectrum of the Koopman operator, constructed using varying dictionary size $N_K$ of 25, 50, 100, and 300. Each plot utilizes 90 initial points to illustrate the impact of increasing the dictionary size on approximating the spectrum of the Koopman operator.

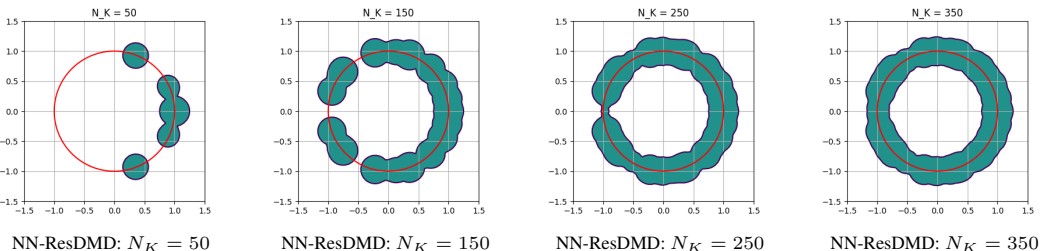

NN-ResDMD: $N_K = 50$    NN-ResDMD: $N_K = 150$    NN-ResDMD: $N_K = 250$    NN-ResDMD: $N_K = 350$

Figure 3: Same example as Figure 2 but with larger data size, using 240 initial points to show the effect of increasing dictionary size on approximating the Koopman operator spectrum.

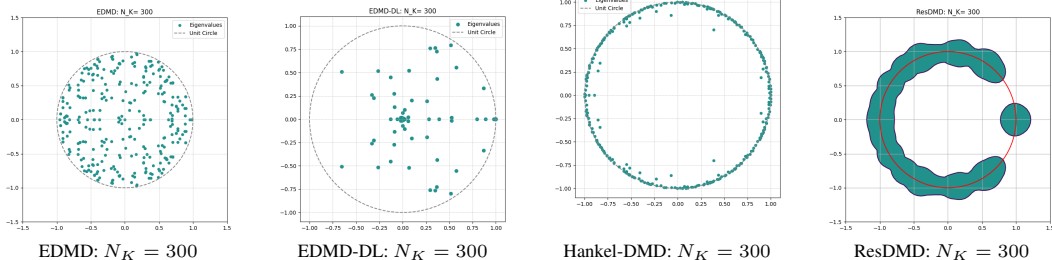

EDMD: $N_K = 300$    EDMD-DL: $N_K = 300$    Hankel-DMD: $N_K = 300$    ResDMD: $N_K = 300$

Figure 4: Comparison with classical methods. The four plots above represent the spectral information obtained from a $300 \times 300$ Koopman matrix, calculated using four methods: EDMD, EDMD with Dictionary Learning (EDMD-DL), Hankel-DMD, and ResDMD. The illustrated eigenvalue spectra of the Koopman operator highlight the differences in results produced by these methods.

## 4.2 TURBULENCE

Recovering spatial patterns is a typical goal of DMD-based methods, especially in fluid dynamics, where Kernel ResDMD has been particularly successful in capturing such patterns and detecting acoustic vibrations in the turbulence system. However, Kernel ResDMD requires careful selection of kernel functions, while NN-ResDMD bypasses this by using neural networks to train observables and compute Koopman modes.

We demonstrate this by applying NN-ResDMD to the turbulence system using the dataset from (Colbrook & Townsend, 2024, Section 6.3). The ground truth in the first plot of Figure 5 represents a high-dimensional pressure field distribution (approximately 30,000 spatial dimensions) around an airfoil, with a clear distinction between the upper and lower surfaces. Technically, we apply truncated Singular Value Decomposition (SVD), select 300 observables, compute Koopman modes, and project them back into the original state space.

In Figure 5, the first Koopman mode estimated by NN-ResDMD with the smallest residual value successfully highlights a clear global spatial separation that aligns with the pattern observed in the original pressure field. This advantage allows the first Koopman mode to directly distinguish spatial features present in the true pressure field, making it a powerful tool for interpreting complex fluid dynamics data. Subsequent Koopman modes also reveal strong acoustic waves that are critical in various aeronautical engineering fields. In contrast, Kernel ResDMD with a generic normalized Gaussian kernel function, as shown in the original work, is unable to produce a Koopman mode similar to the first Koopman mode from NN-ResDMD that clearly distinguishes the pressure field. For comparison, we also plot four Koopman modes computed by Hankel-DMD with a time delay of 5, corresponding to the four smallest residual values, which similarly do not reveal the pressure field patterns as in NN-ResDMD. These results are presented in Appendix Figure 7 and Appenxic A.7.2.

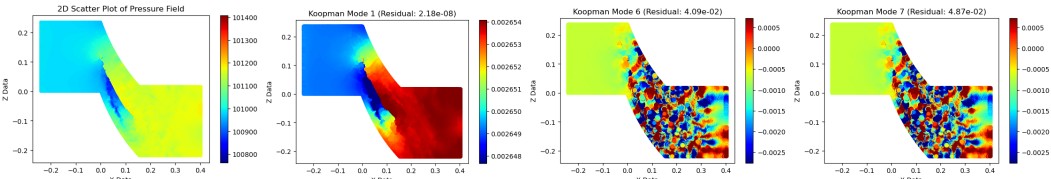

Figure 5: The plots illustrate turbulence detection with Koopman modes computed by 300 observables. The first plot shows a 2D scatter plot of the pressure field, while the other plots display various Koopman modes, each labeled with corresponding residuals. The small residual values in the figures associated with the Koopman modes confirm the estimation accuracy.

### 4.3 Identification of Neural Dynamics in Mice Visual Cortex

Since NN-ResDMD directly minimizes the residuals based on eigenfunctions, its estimated evolution of eigenfunctions over time should ideally capture latent dynamics. To evaluate how effectively NN-ResDMD reveals latent temporal dynamics in real data, we apply it to a dataset of high-dimensional neural signals and demonstrate its advantages over a series of classical methods: the Hankel-DMD, EDMD (combined with RBF basis) and Kernel ResDMD. These methods are selected as representative approaches for handling high-dimensional data.

The dataset is part of the open dataset on mice from the competition "Sensorium 2023" (Turishcheva et al., 2023; 2024). In the experiments, mice viewed natural videos while their neural signals were recorded via calcium imaging in the primary visual cortex, reflecting the activity of thousands of neurons. Here, we focus on the state partitioning of neural signals. Specifically, in each mouse, six video stimuli were repeatedly shown, creating ideal conditions to define brain states. Neural activity during repeated trials with the same stimuli is assumed to reflect the same underlying dynamic system, enabling Koopman decomposition methods to uncover and separate these brain states.

The dataset consists of neural recordings from five mice, each exposed to 6 video stimuli, repeated 9-10 times for a total of around 60 trials. Each recording captures the activity of over 7,000 neurons, with each 10-second video sampled at 50 Hz, resulting in 300 data points per trial.

We applied NN-ResDMD and three classical Koopman decomposition methods (Hankel-DMD, EDMD with RBF basis, and Kernel ResDMD) to these datasets, using different implementations and Koopman subspace dimensions. For NN-ResDMD, we trained dictionaries on all snapshots from each mouse to avoid overfitting, reduced the data to 300 dimensions via SVD, and selected 501 eigenfunctions. The decomposed eigenfunctions are shown in Figure 6A(top), with markers indicating ground truth state separations. For Hankel-DMD, we built a Hankel matrix with a delay of 50, producing 50 eigenfunctions per trial. In EDMD with RBF basis, we used the SVD-truncated 300 basis and 1000 RBF functions, resulting in 1301 eigenfunctions. For Kernel ResDMD, we

used normalized Gaussians as kernel functions, setting the Koopman subspace dimension to 299 eigenfunctions based on Colbrook et al. (2023). See Appendix A.8.4 for method details and Appendix A.9 for dictionary size justification. These eigenfunctions, shown in Figure 6A(bottom), Appendix Figure 9A, and Appendix Figure 10A, are compared to the ground truth trial identities.

The Koopman eigenfunctions represent dynamical features corresponding to the video stimuli. To evaluate their effectiveness, we assess how well eigenfunctions of the same stimuli cluster together, distinguishing them from other states. If the eigenfunctions capture key dynamics related to the stimuli, those from trials with the same video should be separable from others. This turns the problem into a clustering task based on the separability of eigenfunctions across different stimuli. Note that averaged trial differences are even visibly clear for the NN-ResDMD case.

We use Multi-dimensional Scaling (MDS) to visualize how these eigenfunction-based features cluster according to ground truth states. MDS reduces data dimensionality based on similarities, making it ideal for visualizing clustering performance. While UMAP and t-SNE are alternative methods, we show MDS results in 2D space (Figure 6B-E), with similar results for UMAP and t-SNE in the supplementary materials (Appendix Figure 8, Appendix Figure 9C,D and Appendix Figure 10C,D).

The 2D MDS visualization reveals clear separation of features for all 5 mice using NN-ResDMD (Figure 6B), whereas no other method shows clear clustering (Figure 6C-E, Appendix Figure 9B, Appendix Figure 10B). To quantify this clustering, we calculate the Davies-Bouldin index (DBI), a measure of clustering quality that assesses how compact and well-separated the clusters are. A lower DBI indicates more compact clusters that are farther apart from each other, which corresponds to better clustering. The DBI is significantly lower for NN-ResDMD (Figure 6F), suggesting that it captures the latent dynamic structure more effectively than all three other methods. Similar clustering patterns are confirmed with UMAP and t-SNE (Appendix Figure 11).

## 5 CONCLUSION AND FUTURE WORK

Koopman spectral components (eigenpairs) are fundamental to understanding dynamical systems, as they reveal intrinsic patterns and structures underlying complex temporal behavior through a linear framework for analyzing nonlinear dynamics. In this paper, we introduced NN-ResDMD, a method for estimating eigenpairs by minimizing spectral residuals, eliminating ResDMD's need to filter precomputed results. Despite higher computational costs, using neural networks to learn eigenpairs provides a significant advantage by capturing patterns automatically and reducing manual intervention in basis selection. This flexibility is particularly beneficial for high-dimensional systems where traditional methods often struggle. Our experiments demonstrate that NN-ResDMD outperforms classical methods—including EDMD, Hankel-DMD, ResDMD, and their variants—in uncovering critical spatiotemporal characteristics of nonlinear dynamics.

Despite the advantages, NN-ResDMD has several limitations and we discuss the major ones here. First, the neural network structure incurs higher computational costs compared to classical approaches, making it unsuitable for real-time learning tasks (see a brief discussion in Appendix A.5). Second, the deterministic nature of the framework does not account for stochastic aspects of the system, such as those addressed by methods like VAMP Mardt et al. (2018), limiting its applicability to highly noisy data. Additionally, the performance of NN-ResDMD is sensitive to hyperparameter tuning, including network architecture, dictionary size, and training criteria, which can require significant effort to optimize.

Koopman eigenpairs provide unique perspectives into the interpretation of nonlinear dynamical mechanisms, and feedforward neural networks (FNNs) represent an initial step in learning spectral properties directly from data. In recent years, various deep neural network structures have been employed to learn the Koopman representations with different optimization targets other than the spectral residuals (e.g. Lusch et al. (2018); Takeishi et al. (2017); Mardt et al. (2018); Yeung et al. (2019); Otto & Rowley (2019); Azencot et al. (2020); Alford-Lago et al. (2022b); Iwata & Kawahara (2020)(see Appendix section A.4 for a comparison with the VAMP framework). With our approach, we demonstrate that even basic architectures can achieve significant improvements in Koopman operator estimation by using the spectral residual loss. Therefore, future work could focus on refining neural network architectures to enhance the accuracy and efficiency of Koopman eigenpair estimation. One promising direction is the incorporation of Physics-Informed Neural Networks (PINNs)

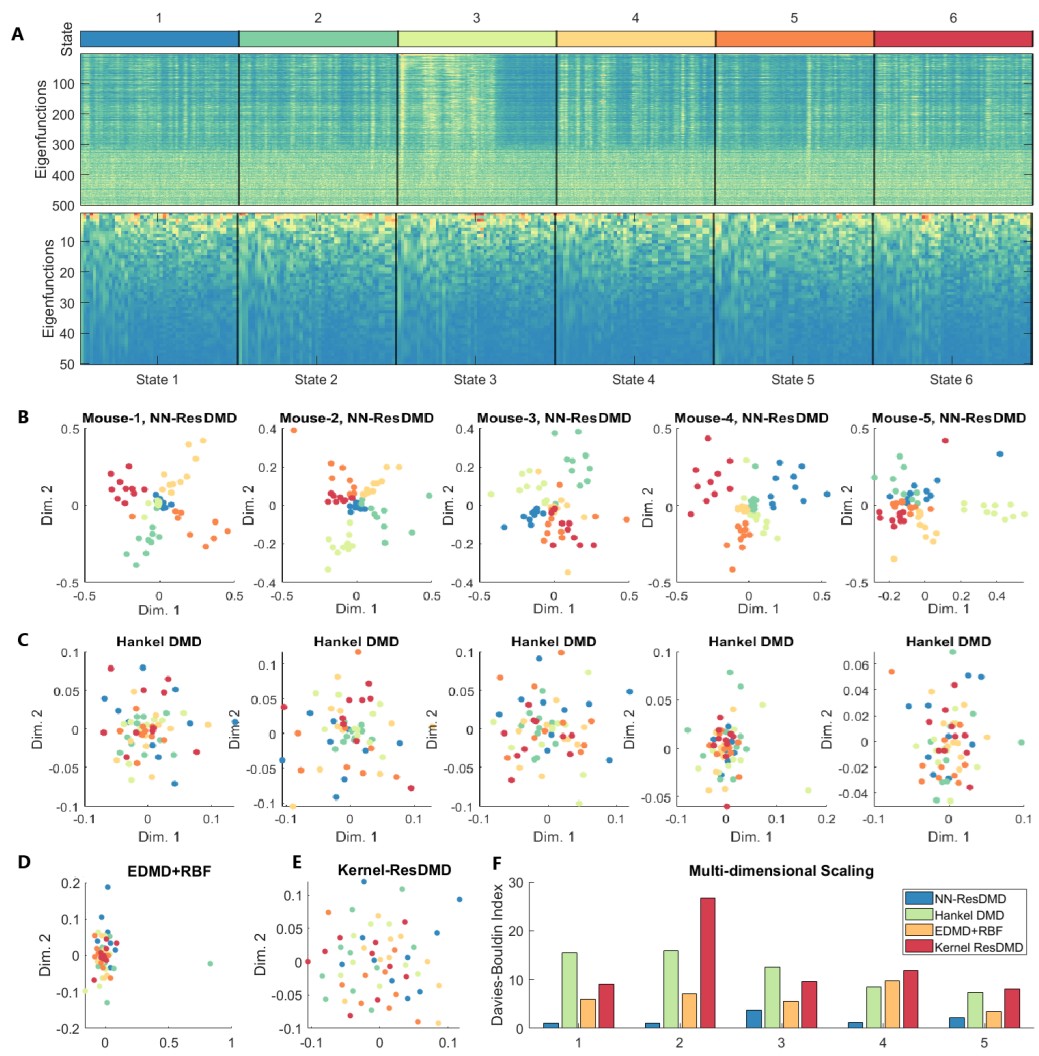

Figure 6: NN-ResDMD outperforms Hankel-DMD in identifying latent dynamic structures in neural signals with a dictionary size of 501. (A) (Top) 500 Koopman eigenfunctions estimated by NN-ResDMD across 6 states characterized by different video stimuli in an example mouse. Each trial contains 300 data points (10s at 50Hz). (Bottom) 50 Koopman eigenfunctions approximated by Hankel-DMD, each 50 points long, reflecting the dimension of the Hankel matrix. (B) 2D representation of Koopman eigenfunctions for all tested mice, computed by NN-ResDMD and reduced via Multidimensional Scaling (MDS). Trials of the same state cluster well. (C) Same as (B) but computed with Hankel-DMD, showing no clear state separation. (D) 2D representation of Koopman eigenfunctions for the first mouse, computed by EDMD with an RBF basis. See Appendix Figure 9 for full results. (E) Same as (D) but computed with Kernel ResDMD. See Appendix Figure 10 for full results. (F) Davies-Bouldin Indices (DBIs) evaluating clustering quality across four methods (NN-ResDMD, Hankel-DMD, EDMD+RBF, and Kernel ResDMD) for five mice. Lower DBI values for NN-ResDMD indicate better clustering.

and Physics-Informed Neural Operators (PINOs), which integrate physical laws directly into the learning process. This integration will ensure that the resulting Koopman eigenfunctions align with known physical constraint, avoid overfitting and faciliatates generalization. Indeed, the integration of PINNs and PINOs with the Koopman framework has the potential to serve as a powerful bridge between data-driven and model-driven approaches, offering enhanced insights into complex systems and enabling more robust temporal evolution predictions.

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

# A APPENDIX

## A.1 CALCULATION STEPS FOR 3.3

Here we are going to show how *squared relative residual* implies (3.1) and then implies (3.3).
Consider $\phi = \boldsymbol{\Psi}\mathbf{v} = \sum_{i=1}^{N_K} \psi_i \mathbf{v}_i$ with $\|\phi\|_2 = 1$, then

$$\frac{\int_\Omega |\mathcal{K}\phi(x) - \lambda\phi(x)|^2 d\mu(x)}{\int_\Omega |\phi(x)|^2 d\mu(x)}$$

$$= \int_\Omega |\mathcal{K}\phi(x) - \lambda\phi(x)|^2 d\mu(x)$$

$$= \langle \mathcal{K}\phi - \lambda\phi, \mathcal{K}\phi - \lambda\phi \rangle_\mu$$

$$= \langle \mathcal{K}\phi, \mathcal{K}\phi \rangle_\mu - \langle \lambda\phi, \mathcal{K}\phi \rangle_\mu - \langle \mathcal{K}\phi, \lambda\phi \rangle_\mu + \langle \lambda\phi, \lambda\phi \rangle_\mu$$

$$= \langle \mathcal{K}\boldsymbol{\Psi}\mathbf{v}, \mathcal{K}\boldsymbol{\Psi}\mathbf{v} \rangle_\mu - \bar{\lambda}\langle \boldsymbol{\Psi}\mathbf{v}, \mathcal{K}\boldsymbol{\Psi}\mathbf{v} \rangle_\mu - \lambda\langle \mathcal{K}\boldsymbol{\Psi}\mathbf{v}, \boldsymbol{\Psi}\mathbf{v} \rangle_\mu + |\lambda|^2 \langle \boldsymbol{\Psi}\mathbf{v}, \boldsymbol{\Psi}\mathbf{v} \rangle_\mu$$

$$= \langle \sum_{i=1}^{N_K} \mathcal{K}\psi_i \mathbf{v}_i, \sum_{j=1}^{N_K} \mathcal{K}\psi_j \mathbf{v}_j \rangle_\mu - \bar{\lambda}\langle \sum_{i=1}^{N_K} \psi_i \mathbf{v}_i, \sum_{j=1}^{N_K} \mathcal{K}\psi_j \mathbf{v}_j \rangle_\mu - \lambda\langle \sum_{i=1}^{N_K} \mathcal{K}\psi_i \mathbf{v}_i, \sum_{j=1}^{N_K} \psi_j \mathbf{v}_j \rangle_\mu + |\lambda|^2 \langle \sum_{i=1}^{N_K} \psi_i \mathbf{v}_i, \sum_{j=1}^{N_K} \psi_j \mathbf{v}_j \rangle_\mu$$

$$= \sum_{i,j=1}^{N_K} \bar{\mathbf{v}}_i \langle \mathcal{K}\psi_i, \mathcal{K}\psi_j \rangle_\mu \mathbf{v}_j - \bar{\lambda} \sum_{i,j=1}^{N_K} \bar{\mathbf{v}}_i \langle \psi_i, \mathcal{K}\psi_j \rangle_\mu \mathbf{v}_j - \lambda \sum_{i,j=1}^{N_K} \bar{\mathbf{v}}_i \langle \mathcal{K}\psi_i, \psi_j \rangle_\mu \mathbf{v}_j + |\lambda|^2 \sum_{i,j=1}^{N_K} \bar{\mathbf{v}}_i \langle \psi_i, \psi_j \rangle_\mu \mathbf{v}_j$$

$$= \sum_{i,j=1}^{N_K} \bar{\mathbf{v}}_i \left[ \langle \mathcal{K}\psi_i, \mathcal{K}\psi_j \rangle_\mu - \bar{\lambda}\langle \psi_i, \mathcal{K}\psi_j \rangle_\mu - \lambda\langle \mathcal{K}\psi_i, \psi_j \rangle_\mu + |\lambda|^2 \langle \psi_i, \psi_j \rangle_\mu \right] \mathbf{v}_j \quad (3.1)$$

$$\approx \sum_{i,j=1}^{N_K} \bar{\mathbf{v}}_i \left[ \frac{1}{m}[\Psi_Y^* \Psi_Y]_{ij} - \bar{\lambda}\frac{1}{m}[\Psi_X^* \Psi_Y]_{ij} - \lambda\frac{1}{m}[\Psi_Y^* \Psi_X]_{ij} + |\lambda|^2 \frac{1}{m}[\Psi_X^* \Psi_X]_{ij} \right] \mathbf{v}_j$$

$$= \frac{1}{m}\mathbf{v}^* \left[ \Psi_Y^* \Psi_Y - \lambda(\Psi_X^* \Psi_Y)^* - \bar{\lambda}\Psi_X^* \Psi_Y + |\lambda|^2 \Psi_X^* \Psi_X \right] \mathbf{v} \quad (3.3)$$

**Remark.** *the inner product above is defined as* $\langle f, g \rangle_\mu = \int_\Omega f^* g \, d\mu(x)$

## A.2 DETAILS FOR DERIVING (3.5)

$$J = \sum_{i=1}^{N_K} \widehat{res}(\lambda_i, \phi_i)^2$$

$$= \sum_{i=1}^{N_K} \frac{1}{m} \mathbf{v}_i^* \left[ \Psi_Y^* \Psi_Y - \lambda_i (\Psi_X^* \Psi_Y)^* - \bar{\lambda}_i \Psi_X^* \Psi_Y + |\lambda_i|^2 \Psi_X^* \Psi_X \right] \mathbf{v}_i$$

$$= \sum_{i=1}^{N_K} \frac{1}{m} \left[ \mathbf{v}_i^* (\Psi_Y^* \Psi_Y) \mathbf{v}_i - \mathbf{v}_i^* (\Psi_X^* \Psi_Y)^* \lambda_i \mathbf{v}_i - \mathbf{v}_i^* \bar{\lambda}_i (\Psi_X^* \Psi_Y) \mathbf{v}_i + \mathbf{v}_i^* K^* (\Psi_X^* \Psi_X) K \mathbf{v}_i \right]$$

$$= \sum_{i=1}^{N_K} \frac{1}{m} \left[ \mathbf{v}_i^* (\Psi_Y^* \Psi_Y) \mathbf{v}_i - \mathbf{v}_i^* (\Psi_X^* \Psi_Y)^* K \mathbf{v}_i - \mathbf{v}_i^* K^* (\Psi_X^* \Psi_Y) \mathbf{v}_i + \mathbf{v}_i^* K^* (\Psi_X^* \Psi_X) K \mathbf{v}_i \right]$$

$$= \sum_{i=1}^{N_K} \frac{1}{m} \Big( \langle \Psi_Y \mathbf{v}_i, \Psi_Y \mathbf{v}_i \rangle - \langle \Psi_Y \mathbf{v}_i, \Psi_X K \mathbf{v}_i \rangle$$

$$\quad - \langle \Psi_X K \mathbf{v}_i, \Psi_Y \mathbf{v}_i \rangle + \langle \Psi_X K \mathbf{v}_i, \Psi_X K \mathbf{v}_i \rangle \Big)$$

$$= \sum_{i=1}^{N_K} \frac{1}{m} \langle \Psi_Y \mathbf{v}_i - \Psi_X K \mathbf{v}_i, \Psi_Y \mathbf{v}_i - \Psi_X K \mathbf{v}_i \rangle$$

$$= \sum_{i=1}^{N_K} \frac{1}{m} \| \Psi_Y \mathbf{v}_i - \Psi_X K \mathbf{v}_i \|_2^2$$

$$= \frac{1}{m} \| (\Psi_Y - \Psi_X K) V \|_F^2.$$

Next, by matrix calculus with denominator layout convention, we try to find minimal of $J$:

$$0 = \frac{dJ}{dK} = \frac{d \operatorname{tr}(J)}{dK} \quad (\text{since } J \text{ is a scalar})$$

$$= \frac{d}{dK} \operatorname{tr} \left( \frac{1}{m} \sum_{i=1}^{N_K} \mathbf{v}_i^* \left[ \Psi_Y^* \Psi_Y - (\Psi_X^* \Psi_Y)^* K \right. \right.$$

$$\left. \left. - K^* (\Psi_X^* \Psi_Y) + K^* (\Psi_X^* \Psi_X) K \right] \mathbf{v}_i \right)$$

$$= \sum_{i=1}^{N_K} \frac{d}{dK} \operatorname{tr} \left( \mathbf{v}_i^* \left[ L - A^* K - K^* A + K^* G K \right] \mathbf{v}_i \right)$$

$$= \sum_{i=1}^{N_K} \frac{d}{dK} \operatorname{tr} \left( \mathbf{v}_i^* L \mathbf{v}_i \right) + \frac{d}{dK} \operatorname{tr} \left( \mathbf{v}_i^* A^* K \mathbf{v}_i \right) + \frac{d}{dK} \operatorname{tr} \left( \mathbf{v}_i^* K^* A \mathbf{v}_i \right) + \frac{d}{dK} \operatorname{tr} \left( \mathbf{v}_i^* K^* G K \mathbf{v}_i \right)$$

$$= \sum_{i=1}^{N_K} -A \mathbf{v}_i \mathbf{v}_i^* - A \mathbf{v}_i \mathbf{v}_i^* + (G + G^*) K \mathbf{v}_i \mathbf{v}_i^*$$

$$= \sum_{i=1}^{N_K} (-2A + 2GK) \mathbf{v}_i \mathbf{v}_i^* \quad (G \text{ is symmetric})$$

where $\operatorname{tr}()$ is trace of a matrix and $G = \Psi_X^* \Psi_X, A = \Psi_X^* \Psi_Y, L = \Psi_Y^* \Psi_Y$.

Since eigenvector $v_i$ is not a zero vector, $v_i v_i^*$ is not a zero matrix. So

$$-2A + 2GK = 0 \Rightarrow K = G^\dagger A.$$

**Remark.** *To solve $\frac{d}{dK} \operatorname{tr} \left( \mathbf{v}_i^* K^* G K \mathbf{v}_i \right)$, we simply rewrite it as*

$$\frac{d}{dK} \operatorname{tr} \left( \mathbf{v}_i^* K^* G K \mathbf{v}_i \right) = \frac{d}{dK} \operatorname{tr} \left( (K \mathbf{v}_i)^* G (K \mathbf{v}_i) \right)$$

A.3 DISCUSSION ON CONVERGENCE

To understand how neural networks enhance NN-ResDMD, it is important to introduce Barron space (Pinkus, 1999; Cybenko, 1989; Haykin, 2009; Barron, 1993). Barron space characterizes functions efficiently approximated by two-layer neural networks, which is central to NN-ResDMD. By leveraging networks that approximate functions within this space, NN-ResDMD can flexibly optimize the dictionary functions for Koopman operator approximation, making it highly effective for complex, high-dimensional systems.

A function $f$ belongs to Barron space $\mathcal{B}$ if it can be represented as:

$$f(x) = \int_\Omega a\sigma(w^T x)\rho(da, dw),$$

where $\sigma$ is the activation function, $w$ is a weight vector, $a$ is a coefficient, and $\rho$ is a probability distribution. The complexity of $f$ is measured by the Barron norm $\|f\|_\mathcal{B}$:

$$\|f\|_\mathcal{B} = \inf_{\rho \in P_f} \left( \int_\Omega |a| \|w\|_1 \rho(da, dw) \right),$$

where $P_f$ is the set of distributions for which $f$ can be represented. This framework provides a basis for analyzing approximation errors in neural networks.

The following theorem (E et al., 2020) discusses the approximation capabilities of two-layer neural networks within this context, establishing a foundation for the subsequent analysis.

**Theorem A.1** (Direct Approximation Theorem, $L^2$-version). *For any $f \in \mathcal{B}$ and $r \in \mathbb{N}$, there exists a two-layer neural network $f_r$ with $r$ neurons $\{(a_i, \mathbf{w}_i)\}$ such that*

$$\|f - f_r\|_{L^2} \lesssim \frac{\|f\|_\mathcal{B}}{\sqrt{r}}.$$

This result implies that the approximation error decreases at a rate of $1/\sqrt{r}$ as the number of neurons $r$ increases, with the constant $\|f\|_\mathcal{B}$ reflecting the complexity of the function $f$ within the Barron space.

Now, consider a Barron space $\mathcal{B}$ which is dense in $L^2(\Omega, \mu)$ and a projected Koopman operator $\mathcal{K}_{N_K} : \mathcal{B}_{N_K} \to L^2(\Omega, \mu)$ where $\mathcal{B}_{N_K} \subseteq \mathcal{B}$ is a $N_K$-dimensional subspace spanned by some dictionary $\mathbf{\Psi} = \{\psi_i\}_{i=1}^{N_K}$. According to Theorem A.1, we can have a well-trained dictionary that almost spans $\mathcal{B}_{N_K}$, i.e., given $\epsilon > 0$, we can always obtain a dictionary $\mathbf{\Psi}_r = \{\psi_{r,i}\}_{i=1}^{N_K}$ such that $\sum_{i=1}^{N_K} \|\psi_{r,i} - \psi_i\|_2^2 < \epsilon$.

A.4 HIGHLIGHTS OF NN-RESDMD COMPARED WITH TYPICAL EXISTING NEURAL NETWORK-BASED KOOPMAN FRAMEWORK

Our NN-ResDMD method takes a fundamentally different approach from existing deep learning methods by building upon the residual-based framework of ResDMD rather than the different Koopman-approximating loss functions following the variational principles of VAMPnets (Tian & Wu, 2021; Wu & Noé, 2020; Mardt et al., 2018) or the deep autoencoder structure in (Lusch et al., 2017). By incorporating spectral residual measures into deep learning and introducing a structured representation that captures dependencies among eigenvalues, we achieve more compact and interpretable models for nonlinear systems with continuous spectra. This approach enables us to directly minimize Koopman spectral approximation errors while avoiding the high-dimensional representations or point-spectrum limitations of previous methods.

If we take the VAMP framework as an example, here are the connections and differences. The proposed loss function and the VAMP score share the goal of optimizing approximations of the Koopman operator's spectral properties, establishing a connection in their ultimate purpose. However, although they both depend on the covariance matrices (in our manuscript Equation 3.2), their methodologies differ significantly. Our residual-based method directly minimizes the spectral approximation error of the Koopman operator and accommodates both point and continuous spectra, while the VAMP score follows a variational framework, maximizing the sum of singular values to

approximate the point spectrum, primarily for stochastic systems (though see an exception in Tian & Wu (2021)). Moreover, while VAMP is specifically designed for Markov processes and requires the Koopman operator to be Hilbert-Schmidt, our approach focuses on deterministic systems and enables a more comprehensive spectral analysis that incorporates continuous spectra. This distinction in scope and methodology highlights how the two frameworks complement each other in addressing different aspects of spectral estimation.

### A.5 DISCUSSION OF COMPUTATION COSTS

Despite the various advantages of the NN-ResDMD framework, one significant limitation is its higher computational cost compared to the original ResDMD and other classical methods.

**Theoretical Perspective:** The NN-ResDMD algorithm's computational demands stem primarily from its iterative optimization process. Each iteration involves a gradient descent update with complexity scaling linearly with both system dimensionality and neural network parameters. Though individual gradient steps are computationally lightweight for standard network architectures, the algorithm's efficiency issue lies in its repeated least-squares optimizations. Below we provide a detailed comparison between computation costs of several methods we used in this paper.

**Comparison over methods:** The computational costs of EDMD, ResDMD, EDMD-DL, Hankel-DMD, and NN-ResDMD vary significantly based on their core computational steps and specific configurations. For a dataset with $m = 10^5$ data points and $N_K = 300$ dictionary functions (for EDMD-based methods), the theoretical complexity and runtime differ across methods. **EDMD** involves least squares and eigenvalue decomposition, with a complexity of $O(N_K^2 m + N_K^3)$, making it the fastest method, and typically requiring only seconds to minutes for computation. **ResDMD** extends EDMD by adding residual evaluation and pseudospectrum computation. The residual evaluation introduces an additional $O(N_K^3)$, and pseudospectrum computation across $n_z$ grid points incurs $O(n_z N_K^3)$, resulting in a total complexity of $O(N_K^2 m + n_z N_K^3)$. This leads to runtimes ranging from minutes to hours, depending on the resolution of the pseudospectrum grid. **EDMD-DL** incorporates dictionary learning through stochastic gradient descent (SGD), where each iteration involves matrix construction ($O(N_K^2 m)$), Koopman matrix computation ($O(N_K^3)$), and neural network forward/backward propagation ($O(d|H|)$, with $d|H|$ representing the total network parameter size). With $k$ SGD iterations, the total complexity becomes $O(k(N_K^2 m + N_K^3 + d|H|))$, leading to runtimes also in the range of minutes to hours depending on $k$. **NN-ResDMD**, which builds on EDMD-DL, shares the same complexity, $O(k(N_K^2 m + N_K^3 + d|H|))$, but includes the explicit use of Koopman matrix eigenvectors and optional pseudospectrum computation, making its runtime slightly longer than EDMD-DL for high-resolution spectral analysis. **Hankel-DMD**, using a time delay embedding dimension $T$, constructs a Hankel matrix ($O(Tm)$), performs singular value decomposition (SVD) ($O(T^2 m)$), and computes the eigenvalues of a reduced $T \times T$ matrix ($O(T^3)$). The total complexity is $O(Tm + T^2 m + T^3)$, and the runtime is heavily influenced by $T$, typically ranging from minutes to hours. While EDMD is computationally the most efficient, ResDMD and Hankel-DMD provide higher precision and robustness in spectral analysis at the expense of increased runtime, and EDMD-DL and NN-ResDMD offer flexibility and accuracy through dictionary learning, with additional SGD iterations and optional pseudospectrum computation contributing to their computational burden.

**Empirical Perspective:** In our experiments, without computing the pseudospectrum, the computational cost of ResDMD typically ranges from seconds to minutes. NN-ResDMD, on the other hand, can require tens of minutes to several hours, depending on factors such as data dimensionality, the number of snapshots, hidden layer configurations, dictionary sizes, and training convergence criteria.

**Trade-off Between Cost and Accuracy:** While NN-ResDMD's additional computational steps introduce higher costs, they enhance the accuracy and robustness of Koopman eigenpair estimation by allowing automatic dictionary learning and minimizing spurious spectral components. This trade-off makes NN-ResDMD particularly valuable in applications where precision is critical. However, its computational demands render it less suitable for real-time or online Koopman model learning tasks.

### A.6 Source code

For reproducibility, the source code will be available at the following anonymous URL: https://anonymous.4open.science/r/ICLR-7305-PROJ. A full version of the codebase will be released upon acceptance of the paper.

### A.7 Hankel-DMD

#### A.7.1 Justification of Using Hankel-DMD as comparison in all experiments

Hankel-DMD operates by constructing a Hankel matrix from time-delayed measurements of the system state, based on Takens' embedding theorem, which states that time-delayed coordinates can reconstruct the state space of dynamical systems. Hankel-DMD also falls within the framework of Extended Dynamic Mode Decomposition (EDMD), as it effectively uses time-delayed states as dictionary functions. This connection introduces convergence conditions specific to time-delay embeddings, differing from those associated with standard EDMD implementations. This makes Hankel-DMD a natural choice for comparison in the pendulum system. Specifically, the method enables a more detailed extraction of the system's modes and dynamics, with theoretical guarantees established in works like (Arbabi & Mezic, 2017), which proved its convergence for ergodic systems.

Practically, the approach involves constructing a large matrix of time-shifted copies of measured data, where the number of delays determines how many past states are considered. This theoretically grounded framework is particularly effective when the system states have good temporal resolution and has shown strong performance in analyzing high-dimensional dynamical systems. Consequently, we also apply Hankel-DMD to the turbulence and neural dynamics experiments to evaluate its effectiveness in these representative high-dimensional settings.

As results, although its performance rivals NN-ResDMD in the simple pendulum system by showing eigenvalues points near the unit circle and containing some polluted eigenvalues, which are close to the ground truth unit circle, we would like to emphasize that it capture the point spectrum and miss the full spectral information. When it comes to high-dimensional systems, it fails to capture key dynamics in higher-dimensional systems, as seen in the later experiment (Section 4.2 and Section 4.3).

#### A.7.2 Application in Turbulence

Here we present the Koopman modes computed by Hankel-DMD for comparison with the NN-ResDMD results. As shown in Figure 7, despite having small residuals, these modes fail to clearly capture the fundamental pressure field structure that was successfully identified by NN-ResDMD's first Koopman mode (see Figure 5). This comparison demonstrates the superior ability of NN-ResDMD to extract physically meaningful patterns from complex fluid systems.

### A.8 Practical details for neural data analysis

#### A.8.1 Dataset Details and Experimental Setup

The dataset utilized in this study is part of the open dataset provided for the 'Sensorium 2023' competition (Turishcheva et al., 2023). The dataset consists of calcium imaging recordings from the primary visual cortex of mice. During the experiments, the mice were presented with natural video stimuli while the activity of thousands of neurons was recorded. The objective of the competition is to predict large-scale neuronal population activity in response to different frames of the stimulus videos, based on the hypothesis that population dynamics in the primary visual cortex, driven by visual stimuli, encode significant information about the dynamics of the videos (Basole et al., 2003; Onat et al., 2011; Hénaff et al., 2021).

#### A.8.2 Task Definition and Rationale

In contrast to the competition's prediction objective, our study focuses on the task of state partitioning of neural signals. While prediction remains feasible, we aim to demonstrate that state partitioning is sufficient to highlight the superiority of NN-ResDMD over a series of other methods in uncovering the latent dynamics of the system. Specifically, in each experiment, a set of six video

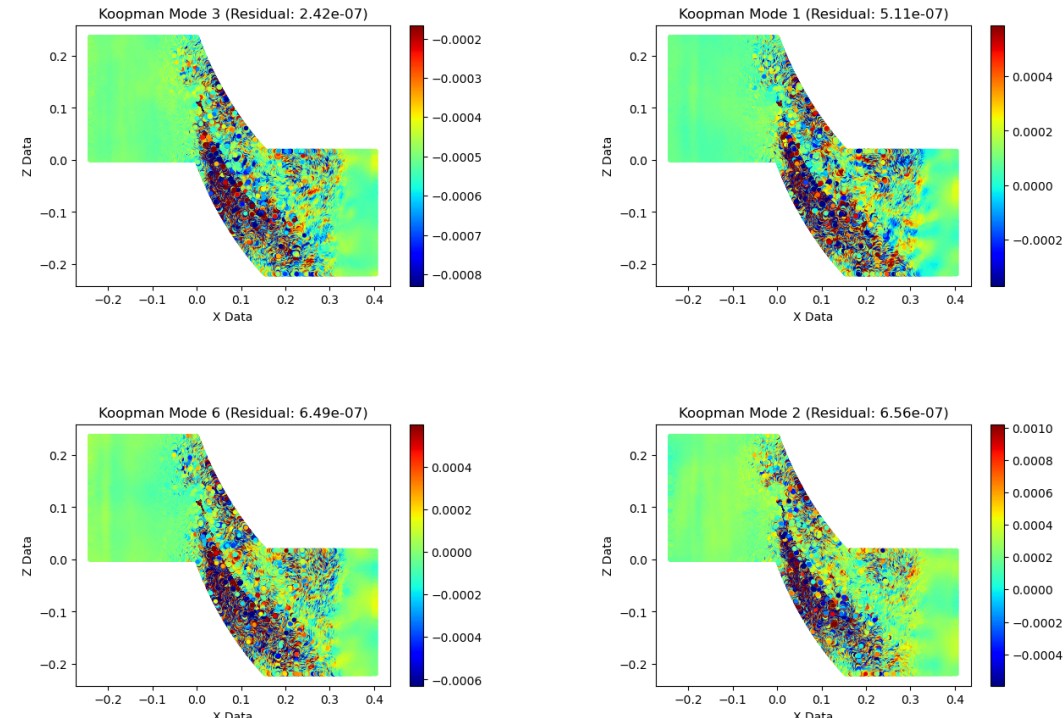

Figure 7: The plots illustrate turbulence detection using the four Koopman modes computed by Hankel-DMD, which are ranked with their corresponding residuals from the smallest.

stimuli was repeatedly presented to each mouse, creating ideal conditions for defining brain states. The recording setup remained consistent for each mouse, ensuring that the neural activities could be interpreted as originating from the same dynamical system, with the primary variable being the input stimulus.

We hypothesize that during repeated trials with identical visual stimuli, the underlying dynamics of the neural system remain consistent. Consequently, the recurrence of the same brain state is expected during these trials. This provides a reliable basis for testing the efficacy of Koopman decomposition methods in uncovering latent dynamics and distinguishing these states.

### A.8.3 DATASET STRUCTURE AND DIMENSIONALITY

The dataset includes neural recordings from five mice, with each mouse responding to six distinct video stimuli, presented in 9-10 repeated trials (resulting in approximately 60 trials in total). Each trial involves recordings of over 7000 neurons. The duration of each video stimulus is 10 seconds, with a sampling rate of 50 Hz, yielding 300 data points (299 snapshots) per trial. Thus, the data to be analyzed consists of a high-dimensional time series with 7000+ observables per snapshot.

### A.8.4 IMPLEMENTATIONS OF NN-RESDMD AND OTHER CLASSICAL METHODS

We compare here four methods: the proposed NN-ResDMD and three classical Koopman decomposition methods for high-dimensional systems: the Hankel-DMD, the EDMD with RBF basis, and the Kernel ResDMD. We applied them to the 5 datasets, although with slightly different implementations and different dimensions of approximated Koopman invariance subspace.

For NN-ResDMD, we train the dictionaries with all the snapshots recorded in each mouse such that the total snapshot number is the product of the snapshot number in one trial and the number of all trials. This is to avoid overfitting with the small snapshot numbers within a trial. The high-dimensional data is first reduced to 300 dimensions with Singular Value Decomposition. The dimension of the

Koopman subspace is chosen to be 601, consisting of 300 trained bases and 301 pre-chosen ones (constant and the first-degree polynomials of the SVD-ed 300 dimensions). The first 501 eigenfunctions sorted by the modulus of eigenvalues are selected to avoid spurious eigenvalues estimation due to noise. One can find the decomposed eigenfunctions in Figure 6A(top), with a marker of the ground truth state separations based on stimulus identity.

For Hankel-DMD, the Koopman eigenfunctions were approximated using the eigenvectors of the Hankel matrix. Specifically, the Hankel matrix was formed as in Equation 53 from Arbabi & Mezic (2017), using all the observables from one trial of each mouse with a delay of 50. Consequently, the snapshot size became 249 times the observable number, and the resulting number of eigenfunctions was 50, each with a length of 50. The Hankel-DMD eigenfunctions for each trial of data are shown in Figure 6A (bottom), alongside the ground truth trial identities for comparison.

For EDMD with RBF basis, the high-dimensional dataset is first reduced to 300 dimensions with SVD. Then RBF basis is calculated with 1000 RBF functions. The choice of the basis number is decided based on classical experiments of using RBF basis to estimate the Koopman operator of Duffing systems (Li et al., 2017).

For Kernel ResDMD, as it is a variant of Kernel EDMD (Kevrekidis et al., 2016), the dimension of the Koopman invariant subspace should corresponds to the sample number (in time). Given the data size to be 300, we have 299 snapshots, resulting in 299 Koopman bases. The detailed calculated is performed for each trial with the program provided in the original ResDMD paper (Colbrook et al., 2023; Colbrook & Townsend, 2024). We chose the kernel function as the commonly-used normalized Gaussian function in the calculation.

The Koopman eigenfunctions from both NN-ResDMD and other methods represent dynamical features corresponding to one of the six video stimuli. To evaluate how well the eigenfunctions capture the latent dynamics, we assess the similarity of the features for trials with the same stimulus and their dissimilarity from those corresponding to different stimuli. Effectively, this makes the problem a clustering task, where the separability of the Koopman eigenfunctions reflects how well they capture the key dynamic components related to the stimuli.

### A.9 CHOICE JUSTIFICATION OF DICTIONARY SIZES

In this section, we provide justifications for the use of different dictionary sizes (i.e., the number of Koopman eigenfunctions) in the aforementioned four methods for the neural dynamics experiment.

First, the high-dimensional data was pre-processed using SVD to reduce its dimensionality to 300. Then, for the four methods:

1. For NN-ResDMD, we selected 300 trained basis functions and 300 first-order monomial basis functions as the dictionary for the 300 reduced observables. This choice ensures the dictionary is rich enough to span the Koopman invariant subspace. Hence, the size of the trained dictionary was set to be at least equal to the original observable size. Then based on the rank of estimated Koopman eigenvalues, we select the dominant 501 eigenfunctions to avoid the eigenfunctions with zero eigenvalues.

2. For Hankel DMD, the number of delays (as dictionary size/number of eigenfunctions) is first constrained by the temporal sample size (i.e., snapshot size) because it cannot exceed the maximum snapshot size. Therefore, it is impossible to choose the same dictionary size as the NN-ResDMD example. Choosing the delay too small will result in an insufficient dictionary size to span the Koopman invariant subspace, and too large will reduce the actual snapshot size to estimate the covariance matrices in the estimation of the Koopman matrix. Therefore, we chose a compromise delay number of 50 that satisfies both needs.

3. For RBF basis, in principle, we can use the same dictionary size. However, our previous experience with a similar dataset and the results of using the RBF basis for the EDMD method all suggest that the performance will be better with more dictionary functions. Therefore, we chose 1000 RBF basis and the original 300 first-order monomial basis as a better condition compared to the same dictionary size with NN-ResDMD.

4. For Kernel ResDMD, the dictionary size is theoretically determined to be the number of snapshots. Therefore, we cannot make the dictionary size consistent with the NN-ResDMD example.

Based on the above justifications, we believe our choices of dictionary sizes are reasonable and ensure a fair comparison across the methods.

### A.9.1 VISUALIZATION AND CLUSTERING PERFORMANCE

To visualize the clustering of high-dimensional Koopman eigenfunctions, we perform dimensionality reduction using Multi-dimensional Scaling (MDS). MDS is particularly useful for visualizing high-dimensional data by preserving pairwise similarities (Kruskal, 1964) (here we use correlation as a measure of similarities). While UMAP (McInnes et al., 2018) and t-SNE (Van der Maaten & Hinton, 2008) are alternative visualization methods, with different emphasis on global-local relationships, we primarily use MDS in this study and provide UMAP and t-SNE results in the supplementary materials (see Appendix Figure 8A, B, Appendix Figure 9C, D and Appendix Figure 10C, D). UMAP in implementation is still correlation-based. For t-SNE estimation we use the perplexity of 15, as a value for optimal separation.

By applying MDS, the high-dimensional eigenfunction-based features are reduced to a low-dimensional space. For illustration, we present the results of reducing the feature space to two dimensions (Figure 6B-E). The NN-ResDMD reduced features for the six types of trials (corresponding to the six video stimuli) are well-separated for all five mice (Figure 6B). In contrast, the Hankel-DMD features show no clear clustering structure (Figure 6C). Similarly, the features produced by EDMD with an RBF basis and Kernel ResDMD do not show clear separability (Figure 6D-E, Appendix Figure 9B-D, Appendix Figure 10B-D).

### A.9.2 CLUSTERING QUALITY METRICS

We further quantified the clustering quality by calculating the Davies-Bouldin Index (DBI) for both Koopman decomposition methods across all mice (Figure 6F). The DBI is designed to assess the compactness of clusters and the separability between them. A lower DBI indicates better clustering performance. NN-ResDMD features yield significantly lower DBI scores compared to other methods, confirming that NN-ResDMD produces more clearly defined clusters corresponding to the ground truth trials. Similar clustering results are observed with UMAP and t-SNE (see Appendix Figure 11), further supporting the superior performance of NN-ResDMD in capturing the latent dynamic structure compared to the other classical methods.

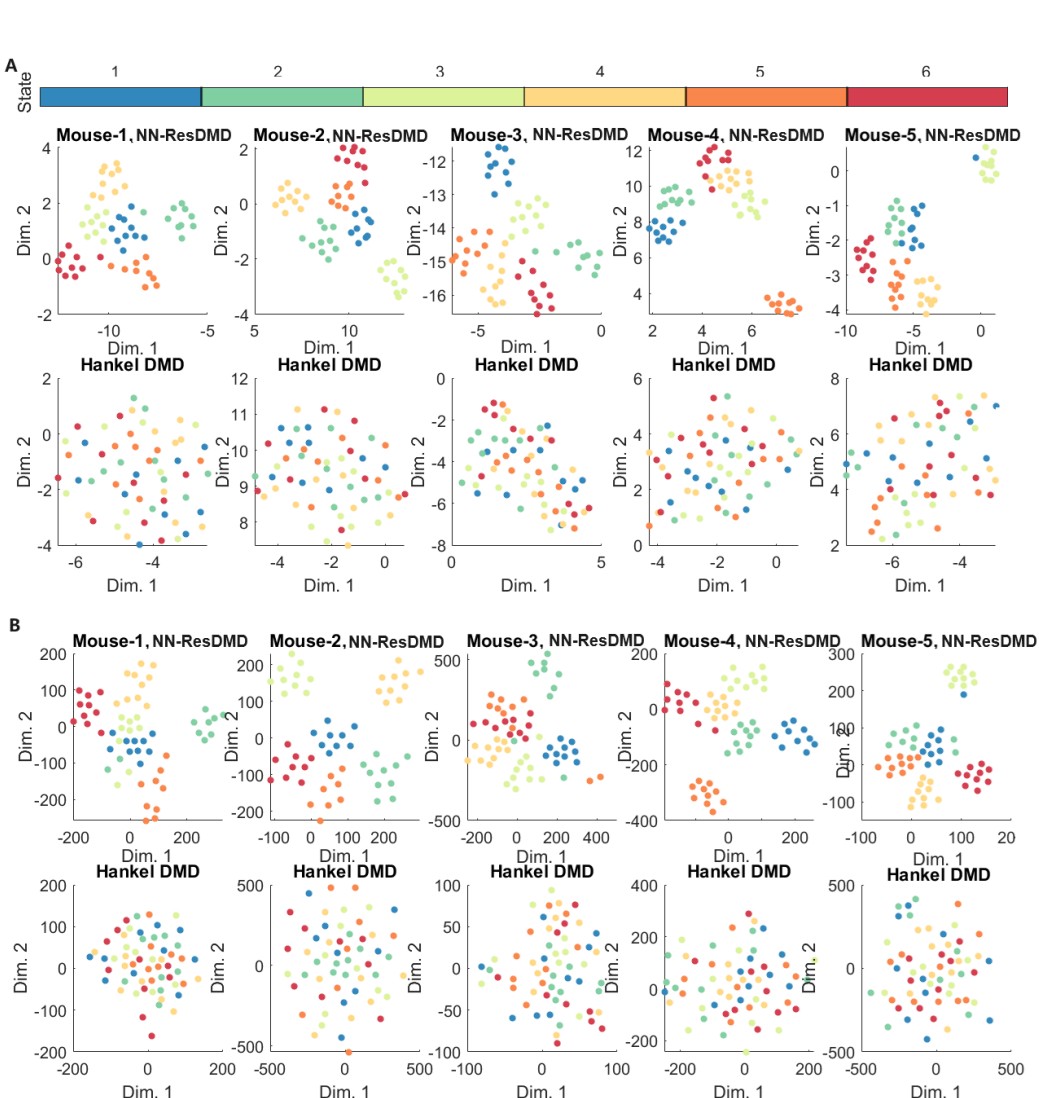

Figure 8: State Partition performance of eigenfunctions for NN-ResDMD and Hankel-DMD in 2D space visualized with UMAP (A) and t-SNE (B).

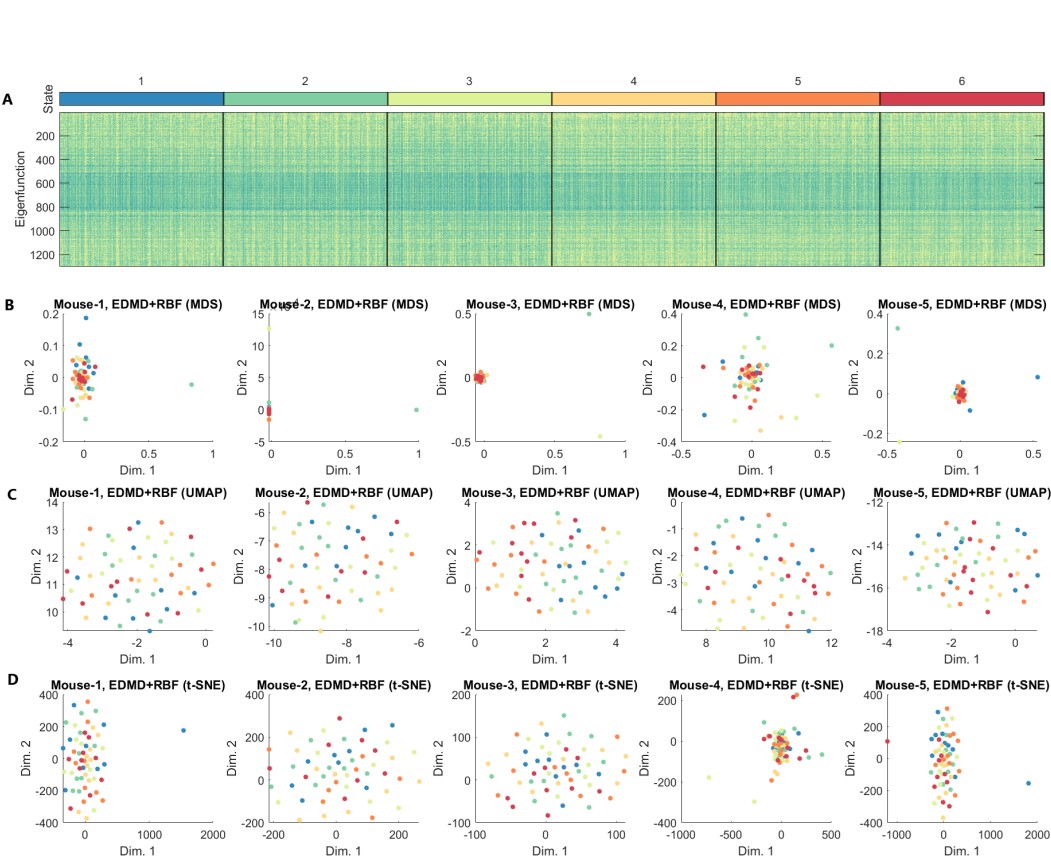

Figure 9: Full results of EDMD with RBF basis. (A) 1301 Koopman eigenfunctions estimated by EDMD with RBF basis in 6 states characterized by 6 different video stimuli in an example mouse. Eigenfunctions in each trial of each state contain 300 data points (10s with a sampling rate of 50Hz). (B) 2-D representation of Koopman eigenfunctions for each trial of all tested mice, calculated by EDMD with RBF basis and reduced by Multidimensional Scaling (MDS). No clear separation of states can be seen from the reduced representation. (C) Same as (B) but visualized with UMAP. No clear separation of states can be seen from the reduced representation. (D) Same as (C) but visualized with t-SNE. No clear separation of states can be seen from the reduced representation.

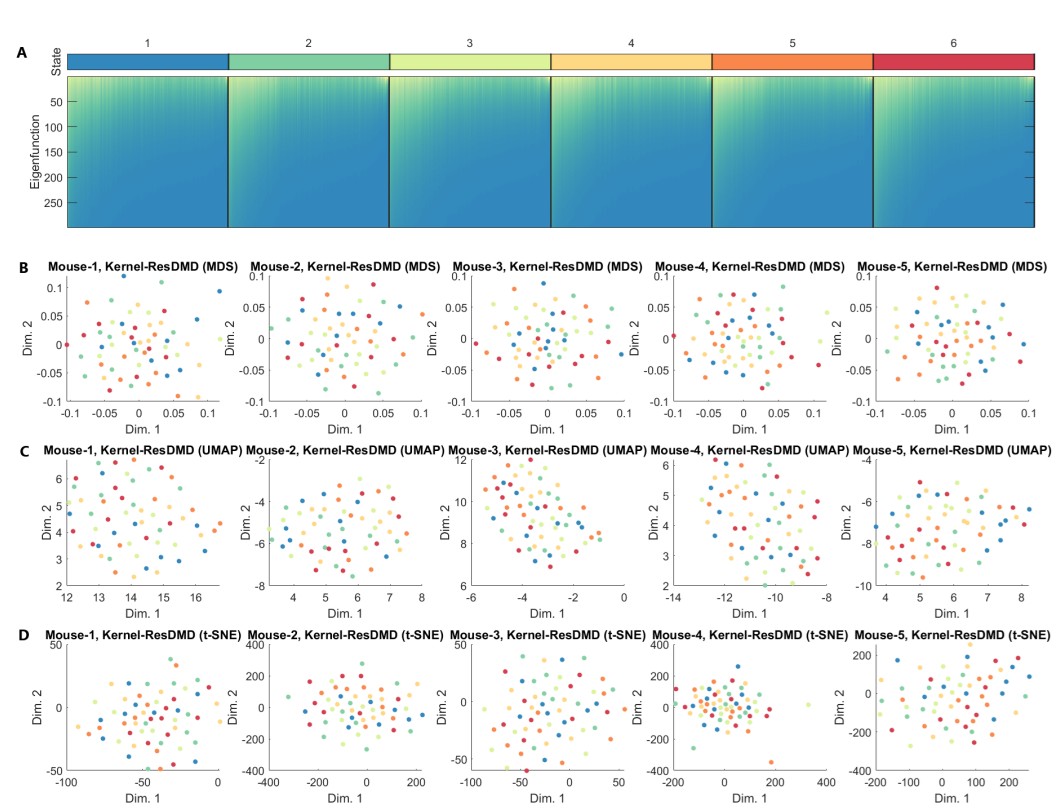

Figure 10: Same as Figure 9 but estimated with Kernel ResDMD, with 299 basis of the Koopman subspace, thus 299 eigenfunctions.

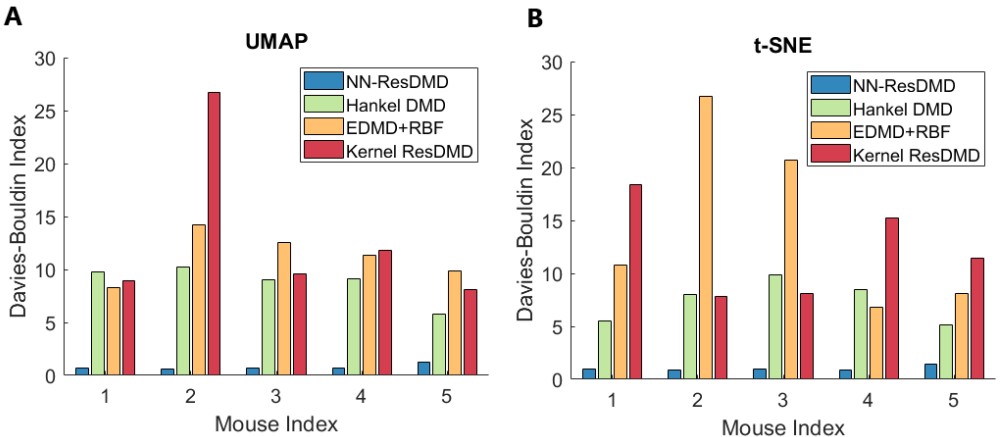

Figure 11: Davies-Bouldin Indices evaluating the clustering performance of dynamical components learned by four methods (NN-ResDMD, Hankel DMD, EDMD+RBF, and Kernel ResDMD) across five mice. Comparisons are shown using UMAP (A) and t-SNE (B).

