# OpenReview forum: "NN-ResDMD: Learning Koopman Representations for Complex Dynamics with Spectral Residuals"
_ICLR.cc/2025/Conference — Submitted to ICLR 2025_

### Official Review · Reviewer_iMTa · 2024-10-20

**Soundness:** 3
**Presentation:** 3
**Contribution:** 2
**Rating:** 6
**Confidence:** 3

**Summary:**

The authors propose Neural Network-ResDMD (NN-ResDMD), where the dictionary functions are automatically selected by neural networks. The network is trained by minimizeing the loss function that is related to the residual of the eigenvalue problem of the Koopman operator. Numerical results are also illustrated to confirm the behavior of the proposed method.

**Strengths:**

ResDMD is a powerful tool to observe the spectra of Koopman operators. The authors combine neural networks and ResDMD to make ResDMD more flexible method. The topic is interesting and relevent to the community. The paper is well-organized and easy to follow.

**Weaknesses:**

The authors insist their proposed method automatically select basis functions, eliminating the need for manual intervention. I understand that in practice, applying neural networks to find proper basis functions is more flexible. However, theoretically, advantages of applying neural networks to estimating Koopman operators and their eigenvalues are not clear for me. They assume there is a basis function $\psi_i$ and construct a neural network that can sufficiently approximate $\psi_i$. Could you clarify what is $\psi_i$? And, what impact on estimating Koopman operators and their eigenvalues is induced by the error of the approximation of $\psi_i$ by using the neural network?

**Questions:**

In Fig. 4, the spectrum of the estiated Koopman operator by NN-ResDMD is distributed on the unit circle. On the other hand, the spectra of the estimated Koopman operators by other methods are distributed also inside the unit circle. Is the dynamical system measure preserving, or did you only focus on the spectra on the unit circle?

Minor comment:
In line 208, do you mean $(G+\sigma I)^{-1}$ instead of $\dagger$ since $G$ is positive-semi definite?

---

> ### Author Response · Authors · 2024-11-20
>
> We sincerely thank the reviewer for the constructive feedback. The questions and doubts mentioned in the **Weakness** and **Question** parts will be answered in the following:
>
> (1) Regarding the question from **Weakness**: The $(\psi_i)_{i=1}^N$ is a set of basis functions that span the Koopman invariant subspace. Common choices are polynomials, Fourier basis, RBF functions, etc. Typically, if you choose different basis functions, the results could vary hugely. However, the optimal choice of basis functions is usually unknown a priori and depends heavily on the specific dynamical system. So, we try to learn the optimal basis functions directly from data by parametrizing the basis functions with neural networks instead of manually selecting them. Next, the error in $\psi_i$ directly impacts the finite-dimensional projection of the Koopman operator. However, our method controls this through spectral residual minimization (Equation 3.3), which ensures the learned basis functions adequately capture the Koopman dynamics. Theoretically, this is justified because the neural network acts as a universal approximator in the Barron space (as discussed in Appendix A.3). We have revised the main text (**lines 83, 88-90, 161-163, highlighted in blue**) to clarify this point.
>
> (2) Regarding the question from **Question**: The pendulum system in Fig. 4 is indeed measure-preserving due to its Hamiltonian nature, which theoretically implies that the whole spectrum including all eigenvalues should lie on the unit circle, i.e., $\|f\|_2 = \|\mathcal{K}f\|_2=\|\lambda f\|_2=|\lambda|\|f\|_2 \Rightarrow |\lambda|=1$. The fact that other methods show eigenvalues inside the unit circle doesn't reflect the true dynamics but rather indicates numerical approximation errors. This highlights a key advantage of NN-ResDMD: By minimizing the spectral residual from ResDMD method directly, NN-ResDMD better preserves the property of the spectrum, which correctly identifies that the whole spectrum(eigenvalues + continuous spectrum) should lie on the unit circle. Traditional methods like EDMD and Hankel-DMD can not handle the continuous spectrum and also generate spurious eigenvalues due to discretization errors, which results in eigenvalues incorrectly appearing inside the unit circle. We have included this property of the pendulum system in **lines 308-309** in the main text (**highlighted in blue**).
>
> (3) For the **minor comment**, the reviewer is correct and we would like to thank him for his helpful correction. We have revised it accordingly in **line 217 (highlighted in blue**) in the main text.

---

> > ### Comment · Reviewer_iMTa · 2024-11-27
> >
> > Thank you for your response. After reading the rebuttal, I'd like to keep my score.

---

### Official Review · Reviewer_GfSg · 2024-10-20

**Soundness:** 2
**Presentation:** 2
**Contribution:** 2
**Rating:** 5
**Confidence:** 3

**Summary:**

This paper introduces a new method, NN-ResDMD, to estimate the spectral components of Koopman operators. It builds upon Residual Dynamic Mode Decomposition (ResDMD) and uses a neural network to identify basis functions instead of manually selecting them.

**Strengths:**

- The description and clarity of the proposed approach is good overall.
- The proposed approach is theoretically justified by the guarantees of the ResDMD method it is built upon.
- The clustering results on the neural dynamics experiment are promising.
- The code for the experiments is already available online.

**Weaknesses:**

**I do not think this paper makes a sufficiently strong contribution**. My understanding is that the proposed approach is simply ResDMD where the basis functions are parametrized by a neural network instead of specified manually, and then iteratively optimized. In my opinion this paper is better suited as a workshop paper in its current form.

The authors use feedforward neural network but the details of the architecture of the feedforward neural network are not provided. The authors **did not investigate the use and choice of other typically better-performing neural network architectures**. I don't think this should be left as future work but should already have been investigated. Similarly, exploring the direction of integrating the proposed approach with PINNs/PINOs would have made the contribution of the paper stronger, instead of leaving it as future work.

There is **no discussion of the computational cost** of the approach, which could be a big drawback of the proposed approach. As far as I understand it, the single evaluation (or few evaluations) of the eigenpairs in ResDMD is replaced by an iterative process where each iteration requires the evaluation of the eigenpairs in NN-ResDMD. This is an expensive part of the algorithm and there are no guarantees for the number of iterations required, so the running time there could be many times larger than for the classical ResDMD. In addition, there is also the cost of the additional optimization steps which can be very large if the neural networks are also large. Overall, the proposed algorithm seems significantly slower than existing approaches, so a trade-off between accuracy and computational time needs to be very carefully discussed theoretically and empirically.

More generally, the **limitations of the algorithm are not discussed**.

**The results of the experiments are not clear**
- The results of the pendulum experiment are not clear. Need to specify more precisely and explicitly what the ground truth is to understand if these are good/bad results. Why are the results of the NN-ResDMD approach shaded areas while all the other methods are displayed using points? The shaded area also has a large radius, so maybe the results are not as good as stated compared to Hankel-DMD for instance for which the points remain close to the unit circle.
- The results of the turbulence experiments are not clear, since there is no ground truth provided. I am not familiar with that experiment so I do not know what the results are supposed to look like. It is unclear to me why the NN-ResDMD results in Figure 5 are considered good, while those in Figure 7 for Hankel-DMD are considered bad.
- The results of the neural dynamics experiments could be made clearer. Why do you choose a different number of eigenfunctions for the different approaches? Is this a fair comparison? Figure 6.a., 9.a. and 10.a. show the decomposed eigenfunctions for the different approaches. There is no ground truth provided, so it is not clear to me what we learn from these plots.

Given that the proposed approach is compared to Hankel-DMD in all the numerical experiments, it would be worth detailing what that approach actually does compared to the other DMD approaches discussed in the paper. Maybe that was the original aim of Appendix A.5 which has been left empty.

The diagram in Figure 1 needs to be cleaner and more "professional"

Make sure to specify the variables over which the optimization is performed in equations (3.4) and (3.5)

**Questions:**

A collection of questions and suggestions have been made in the Weaknesses section.

---

> ### Author Response · Authors · 2024-11-21
>
> Thank you for your thorough and detailed review. We appreciate very much for your time and helpful comments/suggestions and would like to address your concerns in the following aspects:
>
>
> **Contribution**
> Our key innovation lies in transforming spectral residuals into a practical tool for refining Koopman spectral components through iterative application. This process enables more accurate spectral analysis and naturally motivates the use of neural networks for dynamic basis optimization. Unlike the original ResDMD, which passively evaluates pre-computed eigenpairs, our method integrates spectral residuals into an iterative but active filtering framework, directly improving the computation of Koopman spectra and addressing challenges in spectral accuracy that previous methods could not resolve (**See lines 169-177, highlighted in red**).
>
>
> **Neural Network Architecture and Innovation:**
> Our network is a three-layer Feedforward Network and the layer size can be defined manually before every training to adapt to each task.  The activation function for each hidden layer is the tanh function. During training, we use the Adam optimizer for updating the network parameters. We have added the structure details in **lines 247-248 (highlighted in green)** of the main text.
>
>
> While we acknowledge that different neural network architectures could be explored, we deliberately used a simple feedforward network to demonstrate that even basic architectures can achieve significant improvements. The choice of network architecture is secondary to our main contribution of establishing the optimization framework. However, we appreciate the suggestion and will explore more advanced architecture in future work to further enhance performance and robustness. We have clarified these in **lines 481-483 (highlighted in green)**.
>
>
> While we appreciate the reviewer's valuable suggestion to extend our work to PINNs/PINOs, we believe such extensions are beyond the scope of this study for several reasons. First, the integration of PINNs/PINOs requires additional modifications to the framework, including embedding physical constraints directly into the learning process, which involves significant methodological and computational changes. Second, implementing and validating these extensions would require a thorough exploration of appropriate physical constraints and regularization techniques, as well as extensive experiments to ensure fair comparisons. Finally, the primary focus of this work is to establish an optimization framework for Koopman spectrum estimation, and introducing PINNs/PINOs would shift the focus away from our core motivation. Nonetheless, we agree that integrating PINNs/PINOs into the Koopman framework is a promising direction, and we plan to investigate this in future work to further enhance the applicability of our approach.

---

> ### Author Response · Authors · 2024-11-21
>
> **Computational Cost:**
> We appreciate the reviewer's thoughtful comment regarding the computational cost of NN-ResDMD compared to classical ResDMD. While it is true that the computational cost of NN-ResDMD is much higher, we would like to clarify a key misunderstanding in the reviewer's reasoning. In NN-ResDMD, the evaluation of Koopman eigenpairs is not performed explicitly at each iteration of the optimization process. Instead, the loss function is defined based on the Koopman matrix and the dictionary generated at each iteration. The residual is automatically determined by this formulation, which eliminates the need for repeated explicit evaluations of eigenpairs during optimization. Gradient descent is then applied using Adam to minimize this loss, making the optimization process distinct from the classical ResDMD approach, where eigenpair evaluations are directly involved.
>
>
> The NN-ResDMD algorithm's computational demands stem primarily from its iterative optimization process. Each iteration involves a gradient descent update with complexity scaling linearly with both system dimensionality and neural network parameters. Though individual gradient steps are computationally lightweight for standard network architectures, the algorithm's efficiency issue lies in its repeated least-squares optimizations. Compared to standard single least-squares computation as in most numerical algorithms, NN-ResDMD requires multiple iterations to achieve convergence, with stochastic gradient descent methods showing a theoretical $O(1/n)$ convergence rate (See [1]). However, the method's nonlinear optimization nature also presents challenges for establishing concrete convergence bounds and error estimates.
>
>
> Empirically, without computing the pseudospectrum, the computational cost of ResDMD in our experiments typically ranges from seconds to minutes. In contrast, NN-ResDMD can take anywhere from tens of minutes to several hours, depending on factors such as data dimensionality, the number of snapshots, hidden layer configurations, dictionary sizes, and training convergence criteria.
> We acknowledge that NN-ResDMD involves additional computational steps due to its optimization process, particularly when employing large neural networks. However, these additional steps enhance the accuracy and robustness of Koopman eigenpair estimation, making the trade-off worthwhile. Nevertheless, the higher computational demands make NN-ResDMD less suitable for real-time or online Koopman model learning tasks.
>
> We have added discussion on this topic in **lines 267-275** and **Appendix A.5 (highlighted in green)**.
>
> [1] F. Bach and E. Moulines, “Non-strongly-convex smooth stochastic approximation with convergence rate O(1/n),” in Advances in Neural Information Processing Systems (2013) pp. 773–781.
>
>
> **General limitations:**
> Thank you very much for this suggestion and we have included now **a paragraph in the Conclusion section** to demonstrate the limitations of our proposed approach (**highlighted in green**).
>
>
> **Experimental Results:**
> We appreciate your feedback on experimental clarity and address specific points below:
>
> 1. **Pendulum Results:**
> I understand your concern about the clarity and comparison of the pendulum results. The ground truth for this Hamiltonian system is indeed the unit circle, as the continuous spectrum and eigenvalues should lie on it. The shaded areas in the NN-ResDMD results represent the pseudospectrum, which is a key feature of our method that can capture the whole spectrum, unlike other methods that only show eigenvalues as points. While the shaded area may appear broad, this actually demonstrates our method's ability to detect the complete spectrum. This width of the shaded region accounts for computational uncertainties, as exact spectrum computation is computationally impossible. Theoretically, ResDMD guarantees that as this error tolerance approaches zero, the pseudospectrum converges to the true spectrum (the unit circle in this case) without spectral pollution. The Hankel-DMD results, though showing points near the unit circle and containing some polluted eigenvalues, only capture the point spectrum and miss the full spectral information. We revised the content to make this distinction clearer and better explain the theoretical significance of the pseudospectrum visualization in **line 327 highlighted in green**.

---

> ### Author Response · Authors · 2024-11-21
>
> 2. **Turbulence Results:**
> We apologize for any lack of clarity in presenting the turbulence experiment results and appreciate the opportunity to provide more details. The ground truth in the first plot of Figure 5 represents the pressure field distribution around an airfoil, with spatial dimensions of approximately 30,000. This high-dimensional pressure field exhibits a clear spatial separation between the upper and lower surfaces of the airfoil.
>
> Our NN-ResDMD method successfully captures this pressure field structure in its first Koopman mode (the second plot in Figure 5), corresponding to the smallest residual value over all computed eigenpairs. This demonstrates our method’s ability to identify physically meaningful patterns in high-dimensional fluid systems. Specifically, the first Koopman mode accurately reproduces the spatial pattern observed in the ground truth pressure field.
>
> In contrast, while Hankel-DMD is theoretically well-founded and has shown excellent performance in many high-dimensional systems, its results (shown in Figure 7) fail to capture the fundamental pressure field structure. The Koopman modes corresponding to the smallest residuals in Hankel-DMD do not reproduce the clear spatial separation pattern seen in the ground truth.
> We have revised **Section 4.2** to clarify the purpose of the experiment, the ground truth and the interpretations(see **lines 374-386 highlighted in green**). We also added a description of the experiments conducted with Hankel-DMD in Appendix A.7.1(see **lines 924-927 highlighted in green**) and **line 394 in the main text (highlighted in green)** .
>
> 3. **Neural Dynamics Results:**
> We would like to thank the reviewer for raising this point. In addition to the proposed NN-ResDMD method, we also applied three typical Koopman mode decomposition methods which are suitable for high-dimensional datasets to the neural dynamics dataset, which are pre-processed with SVD to a reduced dimension of 300. The reasons of choosing the dictionary size (i.e. the number of Koopman eigenfunctions) are the following:
>
>     (a).For NN-ResDMD, we chose 300 trained basis and 300 first-order monomial basis as the dictionary for the 300 reduced observables because we think we need dictionary rich enough to span the Koopman invariant subspace, thus the size of trained dictionary should be at least the same as the original observable size. Then based on the rank of estimated Koopman eigenvalues, we select the dominant 501 eigenfunctions   to avoid the eigenfunctions with zero eigenvalues.
>
>     (b).For Hankel DMD, the number of delays (as dictionary size/number of eigenfunctions) is first constrained by the temporal sample size (i.e. snapshot size) because it cannot exceed the maximum snapshot size. Therefore, it is impossible to choose the same dictionary size as the NN-ResDMD example. Then choosing the delay too small with result in insufficient dictionary size to span the Koopman invariant subspace, and too large with reduce the actual snapshot size to estimate the covariance matrices in the estimation of Koopman matrix. Therefore, we chose a compromised delay number 50 that satisfies both needs.
>
>     (c).For Hankel DMD, the number of delays (as dictionary size/number of eigenfunctions) is first constrained by the temporal sample size (i.e., snapshot size) because it cannot exceed the maximum snapshot size. Therefore, it is impossible to choose the same dictionary size as the NN-ResDMD example. Choosing the delay too small will result in an insufficient dictionary size to span the Koopman invariant subspace, and too large will reduce the actual snapshot size to estimate the covariance matrices in the estimation of the Koopman matrix. Therefore, we chose a compromise delay number of 50 that satisfies both needs.
>
>     (d).For Kernel ResDMD, the dictionary size is theoretically determined to be the number of snapshots. Therefore, we cannot make the dictionary size consistent with the NN-ResDMD example.
>
> Based on the above justifications, we believe our choices of dictionary sizes are reasonable and ensure a fair comparison across the methods. We have added these details in **Appendix Section A.9** to justify our choices to the readers and mentioned them in **lines 433-434 (highlighted in green)**.
>
>
> While there is no ground truth for how eigenfunctions should behave in these empirical data, the trial labels (e.g., colored bars in Figures 6a, 9a, and 10a) serve as a reference for evaluating method performance. Eigenfunctions estimated by NN-ResDMD exhibit clear differentiation across trial labels even through visual inspection, demonstrating its ability to capture distinct dynamic patterns. This differentiation validates NN-ResDMD’s utility for high-dimensional, complex datasets. We pointed out this differentiation in **lines 441 of the main text (highlighted in green)**.

---

> ### Author Response · Authors · 2024-11-23
>
> **Hankel-DMD**
> Thank you for pointing out this shared methodology. We have added a brief overview to Hankel DMD in **Appendix 7.1 (lines 902-930**, which might also serve as a justification of our baseline method choice, mentioned in the **main text line 303**). We also added a short summary in **lines 301-304** regarding the shared usage of Hankel-DMD.
>
> **Figure 1:**
> Thank you for your suggestion for improving Figure 1. We have updated and included it in the main text.
>
>
> **Optimization Variables:**
> We would like to clarify that the variables in (3.4) and (3.5) are not explicitly defined because they are for theoretical purposes. The explicit definition of optimization variables ($\theta $) that encompass all trainable parameters in the neural network are defined in Equations (3.7) and (3.8). The optimization process follows standard neural network training practices. Nevertheless, the variables in (3.4) and (3.5) align naturally with the later equations, which clarify the parameter optimization process.

---

> > ### Comment · Reviewer_GfSg · 2024-11-26
> >
> > I would like to thank the authors for taking into account the comments and suggestions I made. The clarity and quality of the paper has improved in my opinion, especially the numerical experiments section. This also clarifies how the proposed approach can be advantageous in practice.
> >
> > Small comments:
> >
> > - The comments made by the authors to clarify the pendulum experiments should also be added to the manuscript.
> >
> > - The discussion of computational cost is good, but I believe it could be improved further by adding a comparison to the other methods discussed in the paper (ResDMD, EDMD, HankelDMD), especially by mentioning the additional cost of the operations in NN-ResDMD versus ResDMD.
> >
> >
> >
> >
> > **While I believe the quality and strength of the paper has improved, I still do not think it makes a sufficiently strong contribution.** Possible additions for a stronger contribution are only acknowledged while I believe some of them could or should have been explored in the current paper.
> >
> > **I am updating my overall rating from a 3 to a generous 5, but I would have rated it a 4 if this had been an option.**

---

> > > ### Author Response · Authors · 2024-11-27
> > >
> > > Thank you for your comment and for acknowledging the improvements in the clarity and quality of our manuscript. We appreciate your updated evaluation and your suggestions for further refinement.
> > >
> > > 1. **Pendulum Experiments**:
> > >    The clarifications we provided regarding the pendulum experiments have now been incorporated into the manuscript, as per your suggestion. This ensures that readers can better understand the significance of the results and their practical implications. It is highlighted in lines 325-338 in green color.
> > >
> > > 2. **Computational Cost Comparison**:
> > >    We have added a detailed discussion on the computational costs of NN-ResDMD compared to other methods, including EDMD, EDMD-DL, ResDMD, and Hankel-DMD. This section highlights the additional costs introduced by NN-ResDMD's iterative optimization and pseudospectrum computation, contrasting them with the computational steps and runtime requirements of the other approaches. This addition aims to address your concern about making the comparison more explicit and comprehensive. It is highlighted in lines Appendex A.5 in lines 885-908 in green color.
> > >
> > >
> > > Thank you again for your time and insightful feedback, which have greatly helped us enhance the manuscript. We appreciate your updated rating and hope the revisions align more closely with your expectations.

---

### Official Review · Reviewer_BKBw · 2024-11-01

**Soundness:** 4
**Presentation:** 2
**Contribution:** 2
**Rating:** 5
**Confidence:** 4

**Summary:**

The authors propose NN-ResDMD, a deep learning-based approach that directly estimates Koopman spectral components by minimizing a spectral residual. This method aims to improve the reliability of approximating Koopman spectra in nonlinear dynamical systems by automatically identifying optimal basis functions for the Koopman invariant subspace. The paper presents experiments on physical and biological systems, demonstrating the method's scalability and accuracy for complex dynamics.

My main comment on this paper is that it lacks sufficient innovation or fails to effectively demonstrate its unique contributions. The use of deep learning to estimate Koopman operators has been explored extensively in prior research. For example:

[1] Lusch, B., Kutz, J. N., & Brunton, S. L. (2018). Deep learning for universal linear embeddings of nonlinear dynamics. Nature Communications, 9, Article 4950.
[2] Mardt, A., Pasquali, L., Wu, H., & Noé, F. (2018). VAMPnets for deep learning of molecular kinetics. Nature Communications, 9, Article 5.
[3] Mardt, A., Pasquali, L., Wu, H., & Noé, F. (2020). Deep learning Markov and Koopman models with physical constraints. Proceedings of Machine Learning Research, 107, 451-475.

The squared relative residual proposed in this paper has similarities to the VAMP-E score explored by Wu and Noé in their work on VAMP [4]. The VAMP score framework has served as a basis for many deep learning models, including VAMPnets [2], state-free reversible VAMPnets and GraphVAMPnets. It would be beneficial for the authors to position their spectral residual measure within this established framework, providing a comparative analysis or highlighting any differences in formulation or performance.

[4] Wu, H., & Noé, F. (2020). Variational approach for learning Markov processes from time series data. Journal of Nonlinear Science, 30, 23-66.

**Strengths:**

The proposed NN-ResDMD method offers a deep neural network based approach for estimating Koopman spectral components

**Weaknesses:**

See the Summary

**Questions:**

(1) How does this method differ from existing deep learning approaches for Koopman operator estimation, and what substantial improvements does it offer in terms of robustness, accuracy, or efficiency?

(2) What are the differences and connections between the proposed loss function in this paper and existing evaluation functions like the VAMP score?

---

> ### Author Response · Authors · 2024-11-20
>
> Thank you for your helpful comments and for allowing us to clarify the novelty of NN-ResDMD and its distinctions from existing work.
>
> Regarding our innovation: our key innovation lies in transforming spectral residuals into a practical tool for refining Koopman spectral components through iterative application. This process enables more accurate spectral analysis and naturally motivates the use of neural networks for dynamic basis optimization. Unlike the original ResDMD, which passively evaluates pre-computed eigenpairs, our method integrates spectral residuals into an iterative but active filtering framework, directly improving the computation of Koopman spectra and addressing challenges in spectral accuracy that previous methods could not resolve.(**See lines 169-177, highlighted in red**)
>
>
> Our method takes a fundamentally different approach from existing deep learning methods by building upon the residual-based framework of ResDMD rather than the different Koopman-approximating loss functions following the variational principles of VAMPnets or the deep autoencoder structure in Lusch et al. By incorporating spectral residual measures into deep learning and introducing a structured representation that captures dependencies among eigenvalues, we achieve more compact and interpretable models for nonlinear systems with continuous spectra. This approach enables us to directly minimize Koopman spectral approximation errors while avoiding the high-dimensional representations or point-spectrum limitations of previous methods. These are the main contributions of our framework.
>
> The proposed loss function and the VAMP score share the goal of optimizing approximations of the Koopman operator's spectral properties, establishing a connection in their ultimate purpose. However, although they both depend on the covariance matrices (in our manuscript Equation 3.2), their methodologies differ significantly. Our residual-based method directly minimizes the spectral approximation error of the Koopman operator and accommodates both point and continuous spectra, while the VAMP score follows a variational framework, maximizing the sum of singular values to approximate the point spectrum, primarily for stochastic systems. Moreover, while VAMP is specifically designed for Markov processes and requires the Koopman operator to be Hilbert-Schmidt, our approach focuses on deterministic systems and enables a more comprehensive spectral analysis that incorporates continuous spectra. This distinction in scope and methodology highlights how the two frameworks complement each other in addressing different aspects of spectral estimation.
>
> We have added the above discussion of NN-ResDMD and VAMP in Appendix section A.4 and mentioned it in **line 484 highlighted in cyan**.

---

> > ### Comment · Reviewer_BKBw · 2024-11-24
> >
> > Thank authors for the detailed response. I have also carefully reviewed the comments from other reviewers along with replies. After further consideration, I have gained a better understanding of the value of the work and decided to slightly increase my score. Below are some additional comments:
> >
> > 1) VAMP-like methods applicable to deterministic systems do exist, e.g.,
> > https://doi.org/10.1515/cmam-2020-0130
> > https://link.springer.com/article/10.1007/s00332-019-09574-z
> >
> > 2) The main issue with the proposed method lies in minimizing the residual in Equation (3.1). While this approach can identify some eigenvalues and eigenfunctions, it does not guarantee that the "important" eigenvalues will always be found, nor does it ensure the discovery of N_K "distinct" eigenvalues. I suspect that the optimization results would heavily depend on the initialization, which is likely why authors mentioned the need for a non-trainable basis in Line 246. However, there is no theoretical guarantee that the non-trainable basis will always allow us to identify all the important eigenvalues.

---

> > > ### Author Response · Authors · 2024-11-25
> > >
> > > Thank you for your detailed review and for reconsidering the value of our work. We appreciate your insights, and we are grateful that you took the time to engage deeply with our manuscript and references. Below, we address your additional comments:
> > >
> > > 1. We appreciate the theoretical insights provided by the KVAD framework, which offers an interesting alternative perspective on addressing the limitations of VAMP for deterministic systems. We incorporated relevant discussions and comparisons with this work in our manuscript (lines 865-866, highlighted in blue). While KVAD demonstrates an elegant variational approach using kernel embedding, its numerical experiments are limited to low-dimensional systems (2D Van der Pol oscillator and 3D Lorenz system). In contrast, our NN-ResDMD method has been validated not only on toy models but also on high-dimensional real-world applications, including turbulence systems (~30,000 spatial dimensions) and neural recordings (>7,000 neurons), demonstrating its practical scalability and effectiveness.
> > >
> > > 3. Without considering the challenges inherent to neural network optimization, ResDMD itself provides a solid theoretical foundation to guarantee the recovery of the entire spectrum including eigenvalues. The importance of the non-trainable basis is indeed a crucial aspect of our approach, as it helps prevent scenarios where a poor initialization could result in the spectral residual being zero. We acknowledge that this is fundamentally a machine learning challenge: starting from a suboptimal initialization and converging quickly to a poor local minimum can lead to undesirable eigenvalue estimates. Conversely, achieving a solution closer to the global minimum would significantly enhance the quality of the identified eigenvalues. However, the optimization of neural networks is a broader challenge and falls outside the primary scope of this paper. While it is a promising direction for future work, our focus in this study is on leveraging the ResDMD framework—which has theoretical guarantees for detecting the entire spectrum—and making it more practical and applicable through the integration of neural networks.

---

> > > > ### Comment · Reviewer_BKBw · 2024-11-26
> > > >
> > > > I don't quite understand the statement: "ResDMD itself provides a solid theoretical foundation to guarantee the recovery of the entire spectrum, including eigenvalues." Theoretically, if all the eigenvalues are found, it naturally ensures that the loss function is minimized. However, the reverse is not necessarily true. When the loss function is minimized, it is possible that only part of the eigenvalues have been identified.

---

> > > > > ### Author Response · Authors · 2024-11-26
> > > > >
> > > > > Yes, you raise a valid point. This limitation primarily arises from our use of neural networks, which introduces inherent optimization challenges such as getting trapped in poor local minima due to unfavorable initialization. Neural network implementation can affect the completeness of eigenvalue identification through these optimization issues. In practice, both the neural network initialization and the choice of dictionary size play crucial roles in determining how many eigenvalues can be effectively captured.

---

### Official Review · Reviewer_HKiW · 2024-11-02

**Soundness:** 3
**Presentation:** 3
**Contribution:** 2
**Rating:** 5
**Confidence:** 5

**Summary:**

A method for computing the Koopman spectra of dynamical systems is proposed. It stands on two main ingredients: 1) the use of spectral residual as a loss function, and 2) the use of NNs for constructing observables. Its applications to a pendulum, turbulence, and neural dynamics are presented, and the proposed method is shown to be successful in extracting the Koopman spectra and analyzing the data.

**Strengths:**

- The method is technically reasonable.
- The idea is somewhat new. To my knowledge, using the resDMD objective with neural observables is quite natural but has not been exactly practiced yet.
- The experiments nicely demonstrate the utility of the method.

**Weaknesses:**

(1) Although the work is solid, I do not think the technical contribution is so significant to be included in the ICLR proceedings. The use of neural observables has been known and practiced well in these 8 years or so, and ResDMD has been already well known and discussed recently. The technical contribution of this work inevitably looks incremental.

(2) The rich literature on the use of NNs as DMD observables, other than Li et al. (2017), seems to be overlooked. For example, even restricting the scope to the mere use of NNs for DMD-based analysis (i.e., excluding more applied perspectives such as control), the following papers (and probably more) should be relevant:

- N. Takeishi, Y. Kawahara, T. Yairi: Learning Koopman invariant subspaces for dynamic mode decomposition, Advances in Neural Information Processing Systems 30, 2017, pp. 1130–1140
- B. Lusch, J. N. Kutz, S. L. Brunton: Deep learning for universal linear embeddings of nonlinear dynamics, Nature Communications, vol. 9, no. 1, p. 4950, 2018
- A. Mardt, L. Pasquali, H. Wu, F. Noé: VAMPnetsfor deep learning of molecular kinetics, Nature Communications, vol. 9, no. 1, p. 5, 2018.
- E. Yeung, S. Kundu, N. Hodas: Learning deep neural network representations for Koopman operators of nonlinear dynamical systems, Proceedings of the 2019 American Control Conference, 2019, pp. 4832–4839
- S. E. Otto, C. W. Rowley: Linearly recurrent autoencoder networks for learning dynamics, SIAM Journal on Applied Dynamical Systems, vol. 18, no. 1, pp. 558–593, 2019
- O. Azencot, N. B. Erichson, V. Lin, M. W. Mahoney: Forecasting sequential data using consistent Koopman autoencoders, Proceedings of the 37th International Conference on Machine Learning, 2020, pp. 475–485
- H. Wu, F. Noé: Variational approach for learning Markov processes from time series data, Journal of Nonlinear Science, vol. 30, no. 1, pp.23–66, 2020
- D. J. Alford-Lago, C. W. Curtis, A. T. Ihler, O. Issan: Deep learning enhanced dynamic mode decomposition, Chaos: An Interdisciplinary Journal of Nonlinear Science, vol. 32, no. 3, p. 033116, 2022
- T. Iwata, Y. Kawahara: Neural dynamic mode decomposition for end-to-end modeling of nonlinear dynamics, Journal of Computational Dynamics, vol. 10, no. 2, pp. 268–280, 2023

I do not think all of them should be included in the reference with detail, but at least the existence of such a rich literature should be mentioned to help readers to better understand the context of the research.

Below are relatively minor technical points that I found unclear.

(3) The authors emphasize the "lack of theoretical guarantee of convergence" of EDMD for several times. In what sense is this "lack" supposed? For example, the work by Korda & Mezić:

- M. Korda & I. Mezić: On convergence of extended dynamic mode decomposition to the Koopman operator, Journal of Nonlinear Science, vol. 28, pp. 687–710, 2018

discusses the convergence in some sense.

(4) The proposed method alternates between the gradient-based update of $\theta$ and the least squares solution to get $K$. Is there any insight of this choice? I am asking this because it is also possible to include the least squares within the gradient computation, as done in Takeishi et al. (2017); Otto & Rowley (2019) listed above for example.

(5) In the experiment, does EDMD-DL follow the exactly same configuration as NN-ResDMD except for the loss function? (it should.) Please elaborate on this more clearly to make it easier to assess the benefit particular to the proposed method.

**Questions:**

Although there is no specific question that will surely affect my evaluation, some comments if any on the points listed in the Weaknesses section would be highly helpful.

---

> ### Author Response · Authors · 2024-11-20
>
> Thank you for reviewing our paper and for the insightful comments and suggestions. We welcome further discussion to refine our work. Now let us answer the questions you mentioned:
>
> To Question (1):
> Our key innovation lies in transforming spectral residuals into a practical tool for refining Koopman spectral components through iterative application. This process enables more accurate spectral analysis and naturally motivates the use of neural networks for dynamic basis optimization. Unlike the original ResDMD, which passively evaluates pre-computed eigenpairs, our method integrates spectral residuals into an iterative but active filtering framework, directly improving the computation of Koopman spectra and addressing challenges in spectral accuracy that previous methods could not resolve.(**See lines 169-177, highlighted in red**)
>
> To Question (2):
> We appreciate your suggestions on these relevant works and have incorporated references to better contextualize our research within the broader literature (please check **lines 479-484, highlighted in red**).
>
> To Question (3):
> The key distinction between EDMD and NN-ResDMD in terms of eigenvalue convergence lies in their theoretical approaches and guarantees. In Section 5.3, EDMD proves weak spectral convergence by showing that for any sequence of eigenvalues $\lambda_N$ of $K_N$ with associated normalized eigenfunctions $\phi_N$, there exists a subsequence converging to an eigenvalue-eigenfunction pair of $K$, provided the weak limit of the eigenfunctions is nonzero. This convergence is established under the assumption of a bounded Koopman operator. In contrast, NN-ResDMD, building upon ResDMD's framework, approaches spectral convergence through the minimization of spectral residuals, which provides a more direct and practical way to identify genuine spectral components. This residual-based approach not only requires weaker assumptions (only closed and densely defined operators) but also naturally handles both point and continuous spectra, which effectively filters out spurious eigenvalues. Furthermore, the residual-based convergence in NN-ResDMD also offers a quantifiable measure of approximation quality for each spectral component, which is not available in EDMD's weak convergence framework. We also have added an explanation of these distinctions in the **lines 224-225, 229-230 highlighted in red** of the main text.
>
> To Question (4):
> The alternating optimization strategy in NN-ResDMD separates the least-squares solution for $ K $ from the gradient-based update for $ \theta $, ensuring computational efficiency and numerical stability. This approach guarantees $ K $ is the optimal least-squares solution at each iteration, which allows the optimization to focus entirely on refining $ \theta $. In contrast, methods like Takeishi et al. (2017) and Otto \& Rowley (2019) integrate the least-squares step into the gradient computation, which results in a unified but tightly coupled framework. While this coupling has its merits, it can introduce challenges such as increased complexity in optimization for incorporating prior knowledge or theoretical constraints. By decoupling these steps, NN-ResDMD aligns naturally with spectral residual minimization, facilitating accurate refinement of both point and continuous spectra without additional optimization complexity. This explicit separation also enhances numerical stability and adaptability, which makes NN-ResDMD particularly effective for analyzing complex dynamical systems. We have addressed this in **lines 243-245 highlighted in red** in the main text.
>
> To Question (5):
> Our NN-ResDMD and EDMD-DL share similar algorithmic structures, which naturally arise from the Galerkin approximation framework common to DMD-based methods, notably in the formula for updating matrix $K$. However, the fundamental distinction lies in their theoretical foundations and objectives. EDMD-DL minimizes a Frobenius-norm-based loss function to optimize the least-squares approximation of the Koopman matrix, which follows the traditional EDMD framework. In contrast, NN-ResDMD is built upon ResDMD's theoretical foundation, which minimizes the spectral residual loss and directly evaluates how well the computed eigenpairs satisfy the spectral properties of the Koopman operator. The $K$ matrix update formula, while appearing similar in both methods, serves different purposes: in EDMD-DL it represents the optimal least-squares solution, while in NN-ResDMD it ensures the minimization of spectral residuals. This distinction in the theoretical foundation leads to fundamentally different learning behaviors and better spectral approximation properties in NN-ResDMD.
>
> [1] M. Korda and I. Mezić, "On convergence of extended dynamic mode decomposition to the Koopman operator," Journal of Nonlinear Science, vol. 28, pp. 687–710, 2018.

---

### Author Response · Authors · 2024-11-22

Dear Reviewers,
Thank you for your valuable feedback.  We have addressed each of your individual comments with corresponding color-coded responses. Additionally, we made some modifications (highlighted in purple) after considering all feedback comprehensively. Specifically, the changes in Algorithm 1 now explicitly include the pseudospectrum computation and Koopman matrix estimation, from which eigenvalue-eigenfunction pairs can be directly derived, thus better emphasizing the core advantage of ResDMD methodology. We have also rephrased some text (the purple-highlighted paragraphs) to a more concise version to maintain the 10-page limit. We appreciate your attention to these changes.

---

### Meta-Review · Area_Chair_jcyr · 2024-12-11

**Metareview:**

This paper introduces NN-ResDMD, a method that integrates neural networks into the ResDMD framework to estimate Koopman spectral components for nonlinear dynamical systems. While the approach demonstrates theoretical grounding and some promising experimental results, it faces several critical issues. The paper’s primary limitation is the lack of substantive novelty. The integration of neural networks into Koopman operator estimation builds on well-established methods without providing significant new insights or practical advancements. While the authors argue for improved spectral accuracy, the results lack clear benchmarks and fail to demonstrate a compelling advantage over existing methods. Additionally, scalability concerns, particularly regarding computational costs and runtime, are insufficiently addressed, limiting the method’s applicability to large-scale or real-time systems. Although the authors provided clarifications and additional experiments during the rebuttal, they did not fully resolve concerns about the robustness and interpretability of their approach. The technical contributions appear incremental, and the paper lacks a clear articulation of its unique value relative to prior work. Given these limitations, a decision to Reject is warranted.

**Additional Comments On Reviewer Discussion:**

During the rebuttal period, the authors addressed several key points raised by reviewers. Concerns about the novelty of the method were met with arguments emphasizing the integration of spectral residuals and neural networks for improved Koopman spectral estimation. However, the justification for the approach’s distinctiveness remained insufficient, with reviewers noting that the contributions were largely incremental. Questions regarding computational efficiency and scalability were addressed with additional explanations and runtime comparisons, but these responses highlighted rather than mitigated the method’s computational limitations. The authors also expanded the discussion on related work and clarified technical aspects such as initialization and basis function selection, which improved the presentation but did not substantively strengthen the paper’s core contributions. Overall, while the authors engaged effectively with the feedback and improved the manuscript in clarity and scope, the responses did not fully resolve concerns about the paper’s limited novelty and practical impact. These factors influenced the final decision to Reject.

---

### Decision · Program_Chairs · 2025-01-22

Reject